# iCLOTS: open-source, artificial intelligence-enabled software for analyses of blood cells in microfluidic and microscopy-based assays

Meredith E. Fay[1,2,3,4,5], Oluwamayokun Oshinowo[1,2,3,4,5], Elizabeth Iffrig[1,6], Kirby S. Fibben[1,2,3,4,5], Christina Caruso[2], Scott Hansen[7], Jamie O. Musick[2], José M. Valdez[7], Sally S. Azer[1,2,5], Robert G. Mannino[1,2,3,4,5], Hyoann Choi [1,2,3,4,5], Dan Y. Zhang [4,8], Evelyn K. Williams[1,2,3,4,5], Erica N. Evans[2], Celeste K. Kanne[2], Melissa L. Kemp [1,3,4], Vivien A. Sheehan[2], Marcus A. Carden[9], Carolyn M. Bennett[2], David K. Wood [7] & Wilbur A. Lam [1,2,3,4,5] ✉

While microscopy-based cellular assays, including microfluidics, have significantly advanced over the last several decades, there has not been concurrent development of widely-accessible techniques to analyze time-dependent microscopy data incorporating phenomena such as fluid flow and dynamic cell adhesion. As such, experimentalists typically rely on error-prone and time-consuming manual analysis, resulting in lost resolution and missed opportunities for innovative metrics. We present a user-adaptable toolkit packaged into the open-source, standalone Interactive Cellular assay Labeled Observation and Tracking Software (iCLOTS). We benchmark cell adhesion, single-cell tracking, velocity profile, and multiscale microfluidic-centric applications with blood samples, the prototypical biofluid specimen. Moreover, machine learning algorithms characterize previously imperceptible data groupings from numerical outputs. Free to download/use, iCLOTS addresses a need for a field stymied by a lack of analytical tools for innovative, physiologically-relevant assays of any design, democratizing use of well-validated algorithms for all end-user biomedical researchers who would benefit from advanced computational methods.

In the past several decades, microfluidics, devices designed to apply fluid flow to microscale channels, have increased in use and complexity for wide-ranging applications including lab-on-a-chip assays and clinical diagnostics[1–4]. These devices coupled with optical microscopy provide a means to answer some of biomedical science's most pressing questions, however, the success of these data-rich experiments hinges on detailed analysis and sophisticated interpretation. While excellent image analysis software exists (e.g., ImageJ[5], CellProfiler[6,7], Icy[8], and

[1]The Wallace H. Coulter Department of Biomedical Engineering, Georgia Institute of Technology & Emory University, Atlanta, GA, USA. [2]Department of Pediatrics, Division of Pediatric Hematology/Oncology, Aflac Cancer Center and Blood Disorders Service of Children's Healthcare of Atlanta, Emory University School of Medicine, Atlanta, GA, USA. [3]Winship Cancer Institute of Emory University, Atlanta, GA, USA. [4]Parker H. Petit Institute of Bioengineering and Bioscience, Georgia Institute of Technology, Atlanta, GA, USA. [5]Institute for Electronics and Nanotechnology, Georgia Institute of Technology, Atlanta, GA, USA. [6]Department of Medicine, Division of Pulmonary, Allergy, Critical Care, and Sleep Medicine, Emory University, Atlanta, GA, USA. [7]Department of Biomedical Engineering, University of Minnesota, Minneapolis, MN, USA. [8]The George W. Woodruff School of Mechanical Engineering, Georgia Institute of Technology, Atlanta, GA, USA. [9]Department of Epidemiology, Gillings School of Public Health, University of North Carolina, Chapel Hill, NC, USA. ✉e-mail: wilbur.lam@emory.edu

Ilastik[9]), they are not readily compatible with time-course experiments involving flow. Solutions specialized for microfluidics are typically limited to a specific device design, are proprietary or dependent on other software, and/or rely heavily on coding/scripting[10–13]. To our knowledge, no easy-to-use, adaptable analytical techniques for methods incorporating fluid flow, dynamic adhesion, and/or commercially-available or novel microfluidics have been made widely available to the greater research community. Therefore, researchers often rely on manual analyses, which are tedious, error-prone, and potentially biased, shortcomings that affect reproducibility. This lack of viable image analysis tools results in underutilized data or lost resolution, preventing researchers from fully realizing the potential of their experiments.

Computational methods capable of processing/interpreting large amounts of imaging data efficiently represent a solution. Open-source computer vision, machine learning, and data science libraries (e.g., Trackpy[10], OpenCV[14], Scikit-image[15], Scikit-learn[16], Numpy[17], Pandas[18], Matplotlib[19], and seaborn[20]) implement well-validated, peer-reviewed algorithms[21–24]. However, application requires a level of computational expertise impractical for most researchers. To meet the clear need for microfluidic-centric automated analytical tools, we present iCLOTS, a free and open-source software that adapts these algorithms for specific use with cellular microscopy data, microfluidics, and in vitro assays thereof.

iCLOTS comprises four classes of applications to address the analytical needs of a range of commonly-used microscopy-based assays, with a focus on emerging microfluidic technologies: (1) adhesion assays provide information about morphology and function of cells on biologically-relevant surfaces[25–27]; (2) single-cell tracking assays provide high-throughput measurements of cell dynamics that correlate with cellular phenotype and patho/physiology[28,29], (3) cell suspension fluid flow assays generate velocity profiles from videomicroscopy of transit through channels, which indicate changes in rheology in health and disease[30–32]; and (4) cell accumulation/occlusion assays in microfluidic systems model important pathologic processes such as atherosclerosis, thrombosis, or particle deposition[33–35]. Each image processing application produces single cell- or feature-resolution data describing cell characteristics (e.g., size, fluorescence intensity) and/or movement (e.g., velocity) as observed in any static or microfluidic device design, independent of channel number, size, or dimension. iCLOTS also includes machine learning (ML), a subset of artificial intelligence, clustering algorithms to assist researchers in mathematically characterizing natural groupings, e.g., healthy/clinical sample dichotomies or single-sample subpopulations, within potentially large datasets. Here, we present each application as applied to blood cells, the prototypical biofluid/biospecimen, which are subject to unique requirements and constraints including heterogenous cell types, high cell densities, frequent integration of fluid flow, and increased viscosity. Indeed, enabled by iCLOTS' ease of use and efficient analytical capabilities, we report observations that not only improve our fundamental understanding of inherited bleeding disorders, sickle cell disease (SCD), and sepsis but also have clear clinical relevance as well. In addition, benchmarking our software using blood, with its inherent complexities as a biosample, indicates that our presented tools can be applied to similar experiments using almost every other cell type(s), which in general will be simpler biospecimens and easier to manipulate.

To connect researchers and clinical laboratories that desire the use of newer, more physiologically-relevant assays with computational algorithms that maximize experimental impact, we offer iCLOTS with accessibility and collaboration in mind: iCLOTS is free, is open-source such that all methods are available for inspection or modification by interested users, and is standalone such that users without computational experience fully benefit from all software functionality[36].

Software, extensive documentation, and opportunities for all users to contribute are available at https://www.iCLOTS.org/.

## Results

### Standalone software is designed to balance ease-of-use with maximum functionality

We developed iCLOTS to adapt to assays ranging from standard microscope slide experiments to commercially available flow chambers to novel microfluidic devices (Fig. 1a). Designed as a standalone post-processing software, users can interpret previously-collected data with new resolution or may collect new microscopy data with iCLOTS capabilities in mind (Fig. 1b). The main menu directs users into four main categories of experimental applications (cell adhesion, single cell tracking, velocity profile, and multiscale microfluidic accumulation), an ML interpretation application, and a suite of video preprocessing tools (Fig. 1c). Users interactively adjust parameter values, numerical factors that define how image processing algorithms should be applied, then run the prescribed analysis to produce detailed numerical and graphical data that may be saved for future reference. Tabular numerical data produced by image processing applications can be directly imported back into the software as inputs to ML clustering algorithms (Fig. 1d).

Each application is designed with a similar layout and requires no coding expertise to implement (Fig. 2a; Supplementary Fig. 1, Supplementary movie 1). Detailed help documentation including information on inputs, parameters, outputs, and best practices for image acquisition and analysis are accessible using an on-screen help button and at https://www.iCLOTS.org/. Users select inputs including one or several single image frames, image sequences, or videos (Fig. 2b). The user is guided through a series of windows to describe their data, e.g., selecting a region of interest or relevant color channels, as shown (Fig. 2c). The original image and the image as analyzed displays in the center of the window, with changes in inputs or parameters updating in real-time, allowing users to fit algorithms to their specific set of data (Fig. 2d). Upon running the analysis, users can quickly parse results using automatically-generated graphs. Analyses are completed within seconds to minutes depending on file size and number (Supplementary Table 1). Options to export graphs, images or videos processed to include an index for each feature, and numerical data are presented. Numerical data also contains descriptive statistics (Fig. 2f). Should users need additional interpretation, the ML application mathematically characterizes natural groupings within any number of pooled datasets (Supplementary Fig. 2).

### Single-cell tracking workflows

iCLOTS adapts previously described particle-linking algorithms for use in tracking cells within microfluidic channels[10,21]. Measures such as a maximum size parameter to exclude cell clusters and minimum cell pixel intensity to exclude debris are selected to ensure high-quality data points are captured. All videomicroscopy-based applications in iCLOTS require user input of frames per second imaging rate to convert outputs to time-based values. Single-cell measurements of velocity, size, and optional fluorescence intensity calculated by summing the pixel intensity of the indicated cell region are generated. We demonstrate use of the iCLOTS single-cell tracking application with a previously described, microfluidics-based cell deformability assay designed to measure single-cell mechanical properties[37]. In this assay, a greater cell velocity during transit of a microchannel smaller than the diameter of the cell indicates increased deformability. Mechanical properties of blood cells are important indicators of cell behavior, including red blood cell pathophysiology[38,39], leukemia cell phenotype[40], and cellular response to drug treatment[28]. Cells were perfused through the device at a constant rate using a syringe pump[37]. Here we show application adaptability using a range of blood cells

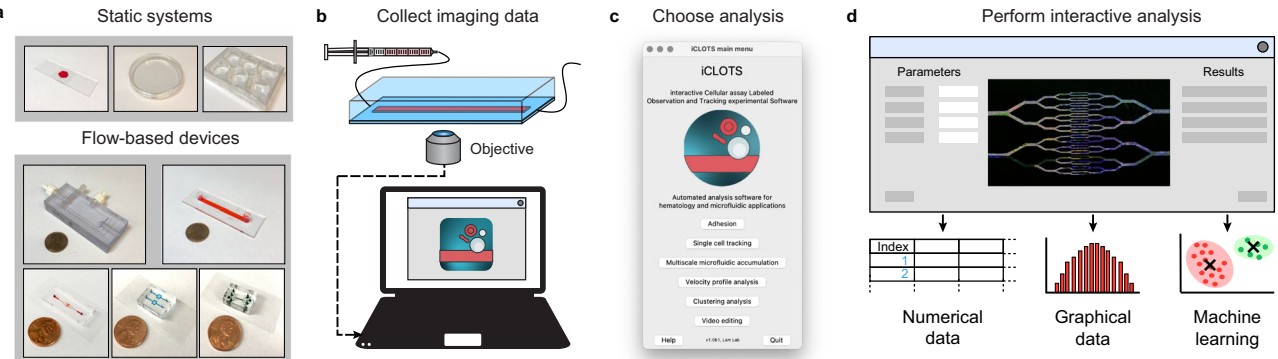

**Fig. 1 | iCLOTS software enables quantification of microscopy data from a wide range of established hematology assays. a** Versatile computational methods adapt to microscopy images and videomicroscopy of cells and cell suspensions obtained using static, standard microscopy assays as well as flow-based systems including traditional flow chambers, commercially-available microfluidic devices, and custom-made microfluidic devices. **b** iCLOTS is designed as a post-processing image analysis software such that users can continue to acquire imaging data using the methods they are accustomed to. This makes iCLOTS suitable for analysis of previously collected data or for new assays planned with iCLOTS' capabilities in mind. **c** Image processing capabilities are separated into four main applications: cell adhesion applications provide single-cell resolution measures of biological functionality, single cell tracking applications provide single-cell resolution measures of cell dynamics and movement including a specialized assay to quantify cellular mechanical properties, velocity profile applications calculate rheological properties of suspensions under flow, and multiscale microfluidic accumulation applications provide insight into potentially pathologic processes such as thrombosis in blood samples. **d** Each application facilitates interactive analysis of a specific experimental workflow. From the microscopy imaging data provided by the user, after image processing algorithms are applied, iCLOTS detects events such as individual cells, patterns of cells, or regions of immunostaining signal, which are then labeled with a number/index on the original and processed imaging files. Numerical output metrics dependent on application type, e.g., cell velocity, area, and/or fluorescence intensity, are calculated for each event and generated as tabular data labeled with the associated index. Additionally, all numerical data is automatically graphed in common formats such as histograms or scatter plots to help users quickly parse the high-dimensional results output by iCLOTS. Should the user need assistance with interpretation of these results, numerical data outputs can be used in post-image analysis applied ML-based clustering algorithms, which assign individual data points with cluster labels suitable for additional methods such as Chi-square analysis.

obtained from clinical samples and cell lines, including red blood cells (RBCs) from patients with SCD (Fig. 3a), RBCs from patients with iron deficiency anemia (Fig. 3b, Supplementary Fig. 3), immortalized white cell lines (Fig. 3c), and reticulocytes from patients with SCD (Fig. 3d). iCLOTS single cell tracking analysis methods are robust, repeatable, and reduce potential for manual error (Supplementary Fig. 3). iCLOTS-produced cell area values correspond with gold-standard clinical blood count (CBC) measurements of mean corpuscular volume (MCV), demonstrating result veracity (Supplementary Fig. 4).

### Machine learning interpretation workflows

iCLOTS uses Python library scikit-learn[16] to implement k-means clustering[41], a specific mathematical model understood to be a robust general-purpose approach to discovering natural groupings within high-dimensional data[42]. Clustering is an unsupervised machine learning technique: pooled single data points (e.g., cells) described by multiple features (e.g., size, circularity, and/or fluorescence intensity) are automatically partitioned into clusters that minimize the differences between shared metrics. A scree plot informs an optimal number of mathematically significant clusters to retain[43]. Outputs include a cluster label for each data point and statistics to assess relative goodness of clustering[44].

### Case study 1: the distinct subpopulation of stiff RBCs characterized by slower velocities within small channels in SCD is also significantly smaller

Decreased RBC deformability is a major manifestation of the genetic condition SCD and may lead to complications associated with microvascular obstruction[45–48]. Measuring deformability characteristics of RBCs from SCD patients at the single-cell level may lead to a better understanding of SCD pathophysiology and prevention of these microvascular occlusive events, reducing incidence of adverse effects such as pain, organ damage, and stroke[46,47]. ML clustering separates all SCD and healthy control RBC data points into an optimal number of

clusters: here, low- and high-velocity groupings (Fig. 3e), in a method akin to flow cytometry with objective boundaries. We find that a higher proportion of SCD RBCs exist in the low-velocity cluster as compared to healthy controls, indicating a significant subset of these cells have increased rigidity as compared to healthy control counterparts (Fig. 3f, Supplementary Fig. 3). In addition to increasing our understanding of SCD RBC variability in the context of pathophysiology, these analyses provide proof-of-concept for subpopulation quantification methods, including a method for objectively defining subpopulation number, which have potential for use in determining clinical reference ranges crucial to new diagnostics.

### Cell suspension velocity profile workflows

Differences in blood or cell suspension rheological properties are caused by a range of diseases and processes, e.g., SCD[30,31], COVID-19[32], or diabetes[49]. Channel flow velocity profiles, a representation of the magnitude of suspension velocity as a function of position, provide valuable insight into sample characteristics including viscosity[50], hematocrit[51], and aggregation[52], but can be challenging to create owing to the many thousands of precise spatial measurements of suspension speed required. Here, image features from high-speed videomicroscopy of cell suspensions under flow, typically pixel intensity patterns representing a grouping of cells, are detected using Shi-Tomasi corner detection[23]. Velocity of features is calculated using Kanade-Lucas-Tomasi optical flow algorithms (Supplementary movie 2)[24]. Users adjust a window size parameter describing the maximum distance from the original feature that the feature may travel in the subsequent video frame. Velocity at each time point and an overall spatial velocity profile is reported. High-throughput quantification is robust, but user-defined detection windows smaller than feature displacement may not accurately capture suspension velocity (Supplementary Fig. 5). All iCLOTS computational methods are developed to adapt to a range of experimental devices, demonstrated here by using a previously described 3-layered microfluidic device that enables initiation of

a    Sample iCLOTS analysis application

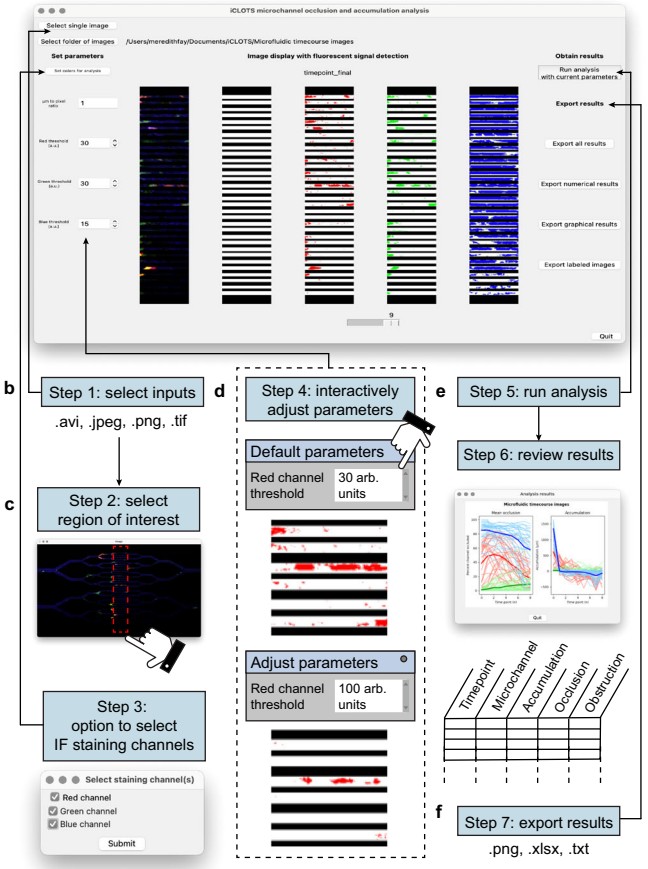

**Fig. 2 | All iCLOTS applications follow a common, easy-to-use interactive format. a** Analysis windows are designed to be intuitively followed from left (inputs) to right (outputs), with the image processing steps as applied displayed in the center. Here, the microchannel analysis application from iCLOTS' suite of multiscale micro-fluidic accumulation tools is shown with whole blood perfused through an in vitro microvasculature-on-a-chip microfluidic model. **b** The user uploads the desired number of microscopy images, time course microscopy series, or videomicroscopy files as inputs. These files are then automatically displayed on the screen. **c** Depending on the application and file type, users are guided through a series of windows facilitating the analyses of their data, such as choosing a region of interest (ROI, shown) or indicating immunofluorescence staining color channels present in a file. iCLOTS applications designed for fluorescence microscopy can accommodate up to three stains in separate channels: here, red indicates CD41+ platelets, green indicates CD45+ white blood cells, and blue indicates the endothelial cell layer. Data in this example is taken at ×100 magnification, scale bars represent 50 μm (left) and 10 μm (right). **d** Parameters, numerical factors that define how image processing algorithms should be applied, are typically simple, e.g., minimum and maximum cell area, or fluores-cence signal threshold, as shown here. All parameters are adjusted interactively from a default value to best match the researcher's specific dataset. In iCLOTS, pixel intensity values are understood to be arbitrary units. Effects of changing parameters are shown in real time to assist in gauging the appropriateness of selected values. **e** A button initiates the finalized analysis with algorithms customized by the selected parameters. Upon completion, graphical results appropriate for the application are automatically displayed, such as line graphs representing quantitative accumulation and occlusion values at each time point for each channel of interest, as seen in this example. **f** Users may export any of the outputs generated by iCLOTS, including tabular data as an Excel file, graphical results as .png images, or the initial imaging dataset as trans-formed by the image processing algorithms and/or labeled with indices.

hypoxic conditions in order to induce sickle hemoglobin polymeriza-tion (Supplementary Fig. 5)[30,31]. As oxygen tension is lowered to 0 mm Hg O$_2$, velocity of SCD RBCs in suspension approaches 0 μm/s (Fig. 4a).

## Case study 2: blood velocity profiles of sepsis patient blood samples are blunted, indicating altered viscosity

Sepsis is a life-threatening infection that leads to inflammatory damage to nearly every organ system[53]. Biochemical abnormalities are known to contribute to sepsis pathophysiology, but are poorly understood[54]. Here we perform a simple microfluidic assay where whole blood samples from patients with sepsis are perfused through a multi-channel microfluidic device at shear rates approximating mean venous shear rates in blood vessels with similar dimensions (Fig. 4b). We find increased overall mean and maximum velocity in sepsis whole blood as compared to a healthy control in a repre-sentative sample (Fig. 4c). Spatial analysis shows profile blunting in sepsis patients that may be explained by the increase in viscosity associated with an acute inflammatory factors like fibrinogen as well as by increased aggregation of red cells in patients with sepsis (Fig. 4d, e)[55]. Use of in vivo capillaroscopy in patients with sepsis have shown a strong correlation between alterations in microvascular velocity profiles and risk of mortality[56]. The temporospatial resolu-tion of these velocity profiles, however, are much less than what iCLOTS provides. Additionally, these alterations measured by in vivo techniques correlate with evidence of endothelial dysfunction[57]. The ability to measure these changes in a tightly controlled in vitro microfluidics platform using iCLOTS permits the exploration for new mechanisms and blood-based biomarkers of endothelial dysfunction in sepsis diagnosis and management.

## Cell adhesion workflows

Experiments in which individual cells adhere to biologically-activated surfaces provide useful information about cellular morphology and physiology (e.g., in platelets[25,26], red blood cells[58], and mixtures of cell types designed to investigate cell-cell interactions[59]). These same cel-lular morphological and functional metrics form the backbone of digital pathology of blood smears[60–62]. Isolated cells were adhered to coated surfaces[25,58]. For all single-cell applications, users choose mini-mum and maximum cell size parameters to reduce noise and exclude cell aggregates, respectively. Individual cells within brightfield micro-scopy images are located as particles represented by image regions with Gaussian-like distributions of pixel brightness[10]. Size and circu-larity metrics are generated as outputs, demonstrated here with het-erogenous cell populations, including small/dense platelets (Fig. 5a) and biconcave-shaped RBCs (Fig. 5b, Supplementary Fig. 6). A separate application for fluorescently stained cells returns additional features including intensity of a secondary stain and texture, a membrane property. Fluorescence images in all iCLOTS applications are seg-mented with a user-chosen binary threshold, a numerical value of arbitrary units where any pixel intensity value above or below the threshold is considered signal to be further quantified or background, respectively. Region property analysis provides morphology metrics of each interconnected region of signal[15]. This application can also distinguish intracellular features such as individual nuclei lobes of neutrophils using signal peak-finding algorithms[15] (Fig. 5c). iCLOTS includes a single cell-resolution protrusion-counting tool (Fig. 5d) based upon Harris corner detection[14,22], demonstrated here to count filopodia-like protrusions within platelets from healthy and clinical samples. Using this application, researchers can objectively apply cri-teria for protrusion detection (Supplementary Fig. 7). Utility of a spe-cialized application to analyze adhesion under flow is demonstrated with neutrophils perfused through a fibronectin-coated microfluidic device (Fig. 5e) Users adjust parameters including a maximum inten-sity value designed to reduce the contribution of debris and a mini-mum number of frames the cell must be present to reduce the contribution of noise. Detected particles representing cells are linked into trajectories used to calculate an adhesion time[10,21]. iCLOTS adhesion assays produce accurate, repeatable, and robust cell

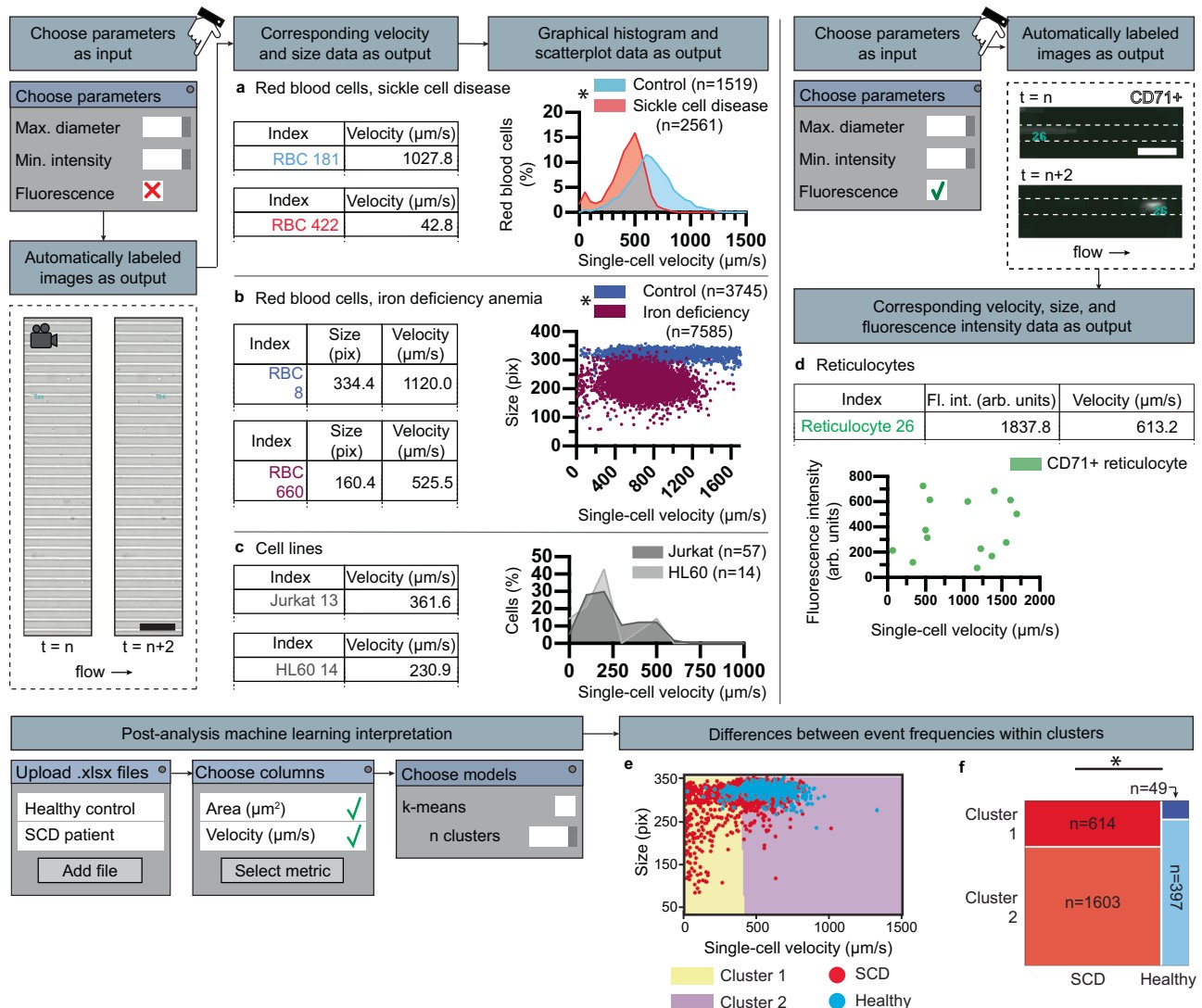

**Fig. 3 | iCLOTS single-cell tracking applications provide high-resolution measurements of velocity relative to cell size and fluorescence intensity.** iCLOTS measures velocity of one or many single cells transiting in any direction(s). We demonstrate use of this application with a specialized microfluidic assay where velocity of a cell transiting a microchannel indicates relative cell stiffness. In all applications, users are guided to adjust input parameters via interactive entry fields. Microscopy data is automatically labeled with image feature (e.g., an individual cell) indices that correspond to a line within an output tabular data sheet. Quantitative velocity, cell size, and optional fluorescence intensity values are calculated for each cell. Histograms of single metrics and scatter plots of multiple metrics are generated. Sample video frames shown taken at ×20 magnification, scale bar represents 100 μm. **a** Dense/dark sickle cell disease patient RBCs ($n = 2561$ RBCs) travel more slowly and thus are less deformable than healthy control RBCs ($n = 1519$ RBCs), including a stiff subpopulation of sickle cell disease patient RBCs with a velocity ranging from 0 to 100 μm/sec. **b** Iron deficiency anemia patient RBCs ($n = 7585$ RBCs) are stiffer and smaller than healthy control RBCs ($n = 3745$ RBCs). *Indicates difference from control ($p < 0.001$ by Mann−Whitney). **c** Heterogenous-intensity cells such as WBCs or leukemia cell lines including Jurkat ($n = 57$ Jurkat cells) and HL-60 cell lines ($n = 14$ HL-60 cells), may also be analyzed using iCLOTS. **d** Optional fluorescence microscopy setting sums the total fluorescence intensity of individual cells, shown here with CD71+ sickle cell disease patient reticulocytes ($n = 14$ reticulocytes). Sample data taken at ×20 magnification, scale bar represents 50 μm. **e** K-means ML clustering algorithms automatically optimize groupings formed from combined SCD and healthy control RBCs into two mathematically defined high- and low-velocity clusters. A scatter plot of chosen metrics with cluster boundaries indicated is generated. **f** Differences between event frequencies within clusters show that more SCD RBCs exist in the low-velocity cluster ($p < 0.0001$ via Chi-squared test). A mosaic plot, a stacked bar chart that shows the percentages of each population within each cluster, is generated. Source data are provided as a Source data file.

---

measurements in a fraction of the time required for manual analysis (Supplementary Figs. 6–8).

**Case study 3: platelet morphology and adhesion on collagen is altered in FLI-1 mutations and Hermansky-Pudlak syndrome**
Adhesion is the first phase of platelet activation after exposure to subendothelial collagen as a result of vascular trauma, and as such, patients with impaired platelet adhesion are at high risk of bleeding[63]. Dysfunction in platelet disorders typically cannot be evaluated with current lab tests due to associated low platelet

count. Specialized adhesion assays assess disorders of primary hemostasis caused by abnormal platelet properties, including in patients with inherited platelet disorders such as FLI-1 mutations and Hermansky-Pudlak syndrome (HPS). FLI-1 mutations are common in patients suffering from platelet dense granule storage pool defects[64]. Additionally, pathological heterozygous mutations in the FLI-1 gene have been commonly shown to also cause macrothrombocytes[65]. HPS is a hereditary disorder that results in platelet dysfunction with prolonged bleeding, among other characteristics. An absence of platelet dense granules has been well

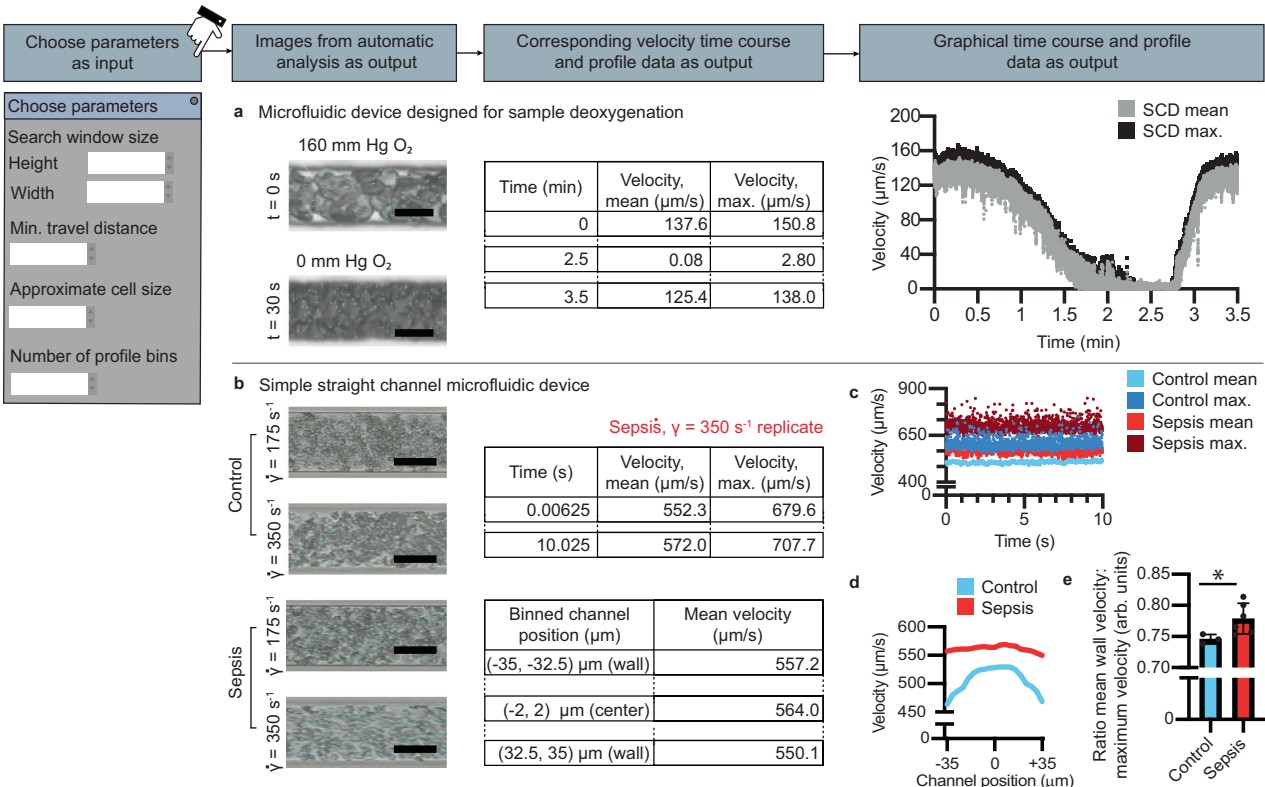

**Fig. 4 | iCLOTS adapts feature finding and tracking algorithms to calculate cell suspension velocity measurements in microfluidic devices. a** Cell suspension velocity applications rely on algorithms that find patterns within images, typically a cluster of cells, and algorithms that track these detected patterns from one frame to the next. To quantify cell suspension velocity, users must adjust window size (a region of interest in which a detected pattern is searched for in the subsequent frame), a minimum distance traveled, and an approximate feature/cell size for best quantification. Trajectories of individual cell patterns are labeled on the provided video data, seen here as cyan lines. Data generated includes a velocity measurement for each cell pattern tracked and mean and maximum velocity for each frame. Mean and maximum velocity measurements of sickle cell patient RBCs in suspension approach zero as oxygen tension is lowered from physiologic oxygen concentrations of 160 mm Hg $O_2$ to deoxygenated conditions of 0 mm Hg $O_2$ ($n = 1$ experiment). Data taken at ×40 magnification, all scale bars represent 10 μm.

**b** Users may also indicate a bin size for automatic generation of channel-wise velocity profiles (representative data from $n = 1$ experiment). taken at ×20 magnification, all scale bars represent 50 μm. **c** Representative time course data shows a consistent mean and maximum velocity over the time course of videos of sepsis patient and healthy control whole blood samples at a shear rate of 350 s$^{-1}$, chosen to recapitulate venous shear rate of a vessel of similar dimensions. **d** Representative profile data shows a blunted velocity profile in sepsis patient whole blood as compared to healthy control whole blood at a shear rate of 350 s$^{-1}$ indicating changes in blood viscosity. **e** Sepsis patient whole blood ($n = 6$ experiments) had a higher ratio of mean wall velocity to frame maximum velocity values as compared to healthy control whole blood ($n = 3$ experiments), indicating a blunted velocity profile, at a shear rate of 350 s$^{-1}$ (*$p = 0.047$ via two-sided Mann–Whitney test). Error bars = standard deviation. Source data are provided as a Source data file.

described in HPS[66]. Using iCLOTS applications, we find differences in adhesion to collagen-coated surfaces, platelet morphology, and phosphatidylserine (PS) exposure in platelet samples from patients with an FLI-1 mutation and HPS as compared to healthy controls (Fig. 5f, Supplementary Fig. 9A). McKneown et al. observed collagen-induced aggregation in patients with HPS[67]. We similarly observed increased spreading and adhesion in platelets obtained from patients with HPS on collagen-coated surfaces. PS on the surface of activated platelets confers a procoagulant surface necessary for hemostasis[68]. Platelets that adhere to collagen transform into rounded structures that expose PS on their surface via Annexin V labeling. ML clustering using k-means algorithms separates all single-platelet data points into optimized clusters describing low and high PS-exposure groupings (Fig. 5g, Supplementary Fig. 9). We find a greater proportion of HPS platelets in the high-PS-exposure cluster as compared to healthy control platelets (Fig. 5h). The single-cell resolution metrics automatically calculated by iCLOTS enable the translational application of platelet function assays designed to provide a deeper understanding of platelet adhesion, with clear implications for new methods of investigation of primary hemostasis in a clinical setting.

## Multiscale microfluidic accumulation workflows

Accumulation, the aggregation or adhesion of cells/biomolecules on biological substrates and/or microvessels, has important implications for multiple diseases[33–35]. Up to three fluorescence microscopy selected image color channels (red, green, and/or blue) from a single image or a time series image sequence are binarized using user-defined thresholds[14]. Pixel values above this threshold are treated as areas of signal to be further quantified into occlusion and accumulation values. iCLOTS' suite of multiscale microfluidic accumulation applications allows users to investigate occlusion on surfaces (Fig. 6a), in potentially complicated microfluidic vessel or channel geometries (Fig. 6b–d), or in small microvessels (Fig. 6e–h). For microfluidic and microvessel applications, a map of potentially complex channel dimensions is created by summing pixel values from images with threshold applied from all channels from all time points. Pixels with an intensity value greater than 0 are considered region(s) of a device. Percent occlusion for each selected color channel is calculated as the signal area divided by total area of a region of the device. Accumulation of cell components on a surface is calculated as the change in signal area measurements between timepoints. In disorders such as atherosclerosis, spatial patterns of accumulation formed in response

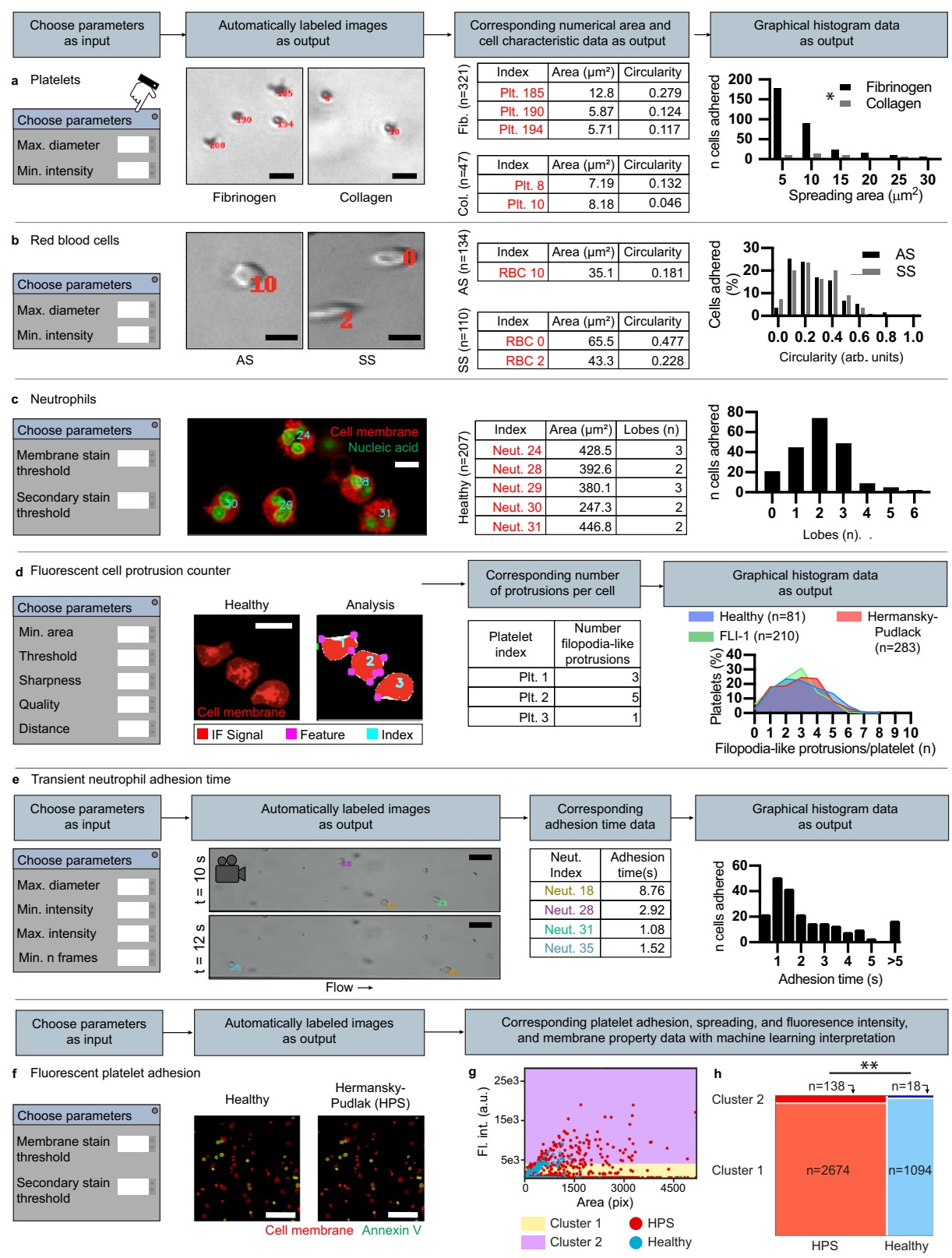

to shear forces generated by changes in vessel geometry take on added significance[33]. In microchannel applications, a spatial percent occlusion is provided at each pixel x-coordinate along the length of the microchannel. These values are additionally summarized into a per-channel occlusion or accumulation (not shown) for each time point. Patterns of occlusion and accumulation persist at a range of thresholds (Supplementary Fig. 10).

## Case study 4: crizanlizumab alters degree and spatial distribution of white blood cell-mediated microvascular occlusion in SCD

Microvascular obstruction in SCD contributes to potentially life-threatening complications including pain crises, organ dysfunction, and stroke[45,69]. P-selectin mediates binding of RBCs and WBCs to the endothelial vessel wall activated by inflammation of trauma, which

**Fig. 5 | iCLOTS cell adhesion applications provide indexed single-cell measurements of biological functionality.** After adjustment of relevant parameters, iCLOTS calculates numerical area and circularity values for individual cells within brightfield microscopy images, including **a** dark/dense platelets adhered on fibrinogen-coated surfaces ($n = 231$ platelets) and collagen-coated surfaces ($n = 47$ platelets) and **b** biconcave RBCs from patients with sickle cell trait (AS) genotypes ($n = 134$ RBCs) and sickle cell disease (SS) genotypes ($n = 110$ RBCs). Platelets adhered to fibrinogen surfaces spread less than those adhered to collagen (*$p < 0.0001$ via two-sided Mann–Whitney test). Data taken at ×30 magnification, scale bars represent 10 μm (**a** and **b**). To analyze fluorescence microscopy imaging data the user indicates pixel value thresholds for membrane and optional secondary stains and additional texture and staining intensity metrics are calculated per-cell. **c** Users may **c**ount regions of a secondary stain, here the number of nuclei lobes in neutrophils ($n = 207$ neutrophils). Data taken at ×20 magnification, scale bar represents 10 μm. **d** A cell protrusion characterization application calculates the number of filopodia-like protrusions present in an individual cell using

additional application-specific parameters designed to apply objective requirement criteria. Data taken at ×20 magnification, scale bars represent 10 μm. **e** Transient adhesion time of individual cells to a biochemically-coated surface is calculated from videomicroscopy data, shown here with neutrophils ($n = 1$ experiment, $n = 185$ neutrophils) in a fibronectin-coated channel. Users adjust additional parameters designed to ensure veracity of returned data points. Data taken at ×20 magnification, scale bar represents 50 μm. **f** Analysis of fluorescence microscopy data of platelets reveals differences in the density of adhered platelets, the spreading area of individual platelets, and phosphatidylserine exposure in individual platelets from healthy controls and a Hermansky-Pudlak Syndrome patient. Data taken at ×40 magnification, scale bars represent 200 μm. **g** K-means ML analysis separates combined healthy ($n = 1112$) and HPS ($n = 2674$) platelets into two groups representing low- and high-PS exposure ($n = 1$ experiment). **h** The proportion of cells in cluster 2, the high-PS cluster, is greater in HPS samples than in healthy controls (**$p < 0.0001$, Chi-squared test). Source data are provided as a Source data file.

in proinflammatory states such as SCD leads to adherent RBC/WBC clusters[70]. Crizanlizumab, a P-selectin inhibitor, has been shown to mitigate microvascular vaso-occlusion in SCD, thereby reducing pain crises[71,72]. Whole blood samples from patients with SCD were perfused through an endothelialized microvasculature-on-a-chip device prepared as described previously[73]. The iCLOTS accumulation application automatically generates a map of complex microfluidic device dimensions (Fig. 6b). We expectedly observed that both CD41+ platelets and CD45+ WBCs within whole SCD blood treated with crizanlizumab occlude microfluidic channels less than untreated SCD blood samples (Fig. 6c) at a lowered percentage rate similar to completed clinical trials (Supplementary Fig. 11)[70]. However, we also observed that the occlusion process is unstable and that accumulation patterns of CD41+ platelets and CD45+ WBCs change over time (Fig. 6d). ML analysis quantifies differences in spatial relationships of occlusion signal within microfluidic microvessels (Fig. 6g, h, Supplementary Fig. 12). WBCs have long been understood to contribute to SCD pathophysiology[74,75]. We find that in SCD whole blood samples not treated with crizanlizumab, at early timepoints CD45+ WBCs primarily occlude ends of channels, but this effect is reversed when SCD blood is treated with crizanlizumab (Fig. 6i, Supplementary Fig. 12). Here, iCLOTS' ability to simultaneously monitor multiple components of a cell suspension with high spatial resolution facilitates greater understanding of SCD pharmacological mechanisms via observed changes in the contribution of WBCs to microvessel occlusion.

iCLOTS applications are designed to close the existing gap between experimental and analytical microfluidic methods by translating information-rich microscopy data into a series of detailed, quantitative results describing a range of cell characteristics and behaviors (Table 1).

## Discussion

Microfluidic technology has progressed significantly in the last several decades, but there has not been a concurrent push to develop automated imaging and analytical techniques for those new experimental methods. ImageJ[5], CellProfiler[6,7], Icy[8], and Ilastik[9] provide excellent, user-friendly implementations of image segmentation, applied computer vision, and/or cellular morphology quantification but do not fill the gap for time-dependent, fluid flow-based and microfluidics-based, experimentally-driven applications. Other specialized open-source tools[10–13] and industry-based proprietary software typically require additional software, onerous complex calculations and/or coding/scripting on the user's part, or are specific to a certain microfluidic device, which excludes many researchers and clinical laboratories from accessing and using those tools. Conversely, iCLOTS algorithms and applications have all been designed to adapt to any microfluidic device or static system, independent of channel number, size, or

dimensions, and is available as a standalone, easy-to-use product to any and all biomedical researchers. iCLOTS is a free software specialized for fluid flow-based cellular microscopy experiments performed using any microfluidic that guides users end-to-end through data analysis and interpretation in one simple, standalone package via a unique combination of validated image processing, feature quantification, and ML algorithms. Should users need more advanced AI-based feature classification from software such as Ilastik, the modular design of iCLOTS permits users to upload segmentation outputs as an initial dataset for further quantification. Thus, instead of being a competing software to established bioimage analysis tools, we see iCLOTS as a complementary solution that focuses on the needs of specific users and experiment types.

As the prototypical biofluid biospecimen, we have found blood cells and related cell suspensions to be the most useful test case to benchmark iCLOTS for dense numbers of different cellular subpopulations, potentially under physiological flow conditions. As such, clearing a high bar with blood samples indicates that iCLOTS is also compatible with other biofluid/cell samples, which typically will be simpler than blood. iCLOTS was designed around large data sets from multiple research groups and has been used to recreate key findings from previously published studies[76–79] (Supplementary Fig. 13) in order to verify that the algorithms applied work consistently for a variety of experimental set ups, imaging parameters, and additional cell or multicellular structure types. However, analysis success depends on the quality of imaging data presented (Supplementary Fig. 14). Some level of noise or spurious features is to be expected with all high-throughput data analysis. iCLOTS is designed to label all analyzed imaging data with single-cell indices corresponding to numerical data, allowing users to assess if outliers are valid data points. If data is unacceptably noisy, it may still be less labor-intensive to correct computationally produced data than to perform manual analysis, which may be error-prone itself. Manual analysis is also prone to bias that may contribute to reproducibility issues, especially if data is analyzed in an unblinded fashion, in a way that iCLOTS results are not. This potential to reduce bias and apply algorithms without need for computational expertise makes iCLOTS especially well-suited for use in clinical laboratories, where making key medical treatment decisions relies on robust, reproducible results. In this way, iCLOTS enables a semblance of standardization of data analysis for cellular microscopy data which is needed if these assays are to be used as clinical diagnostics and the regulatory processes thereof. Clustering techniques are well-suited to exploring distinguishing features between known populations and to finding new, previously imperceptible groupings within a single population. However, metrics describing populations of cells typically follow Gaussian distributions which may have significant overlap. In an effort to guide users through the analysis and

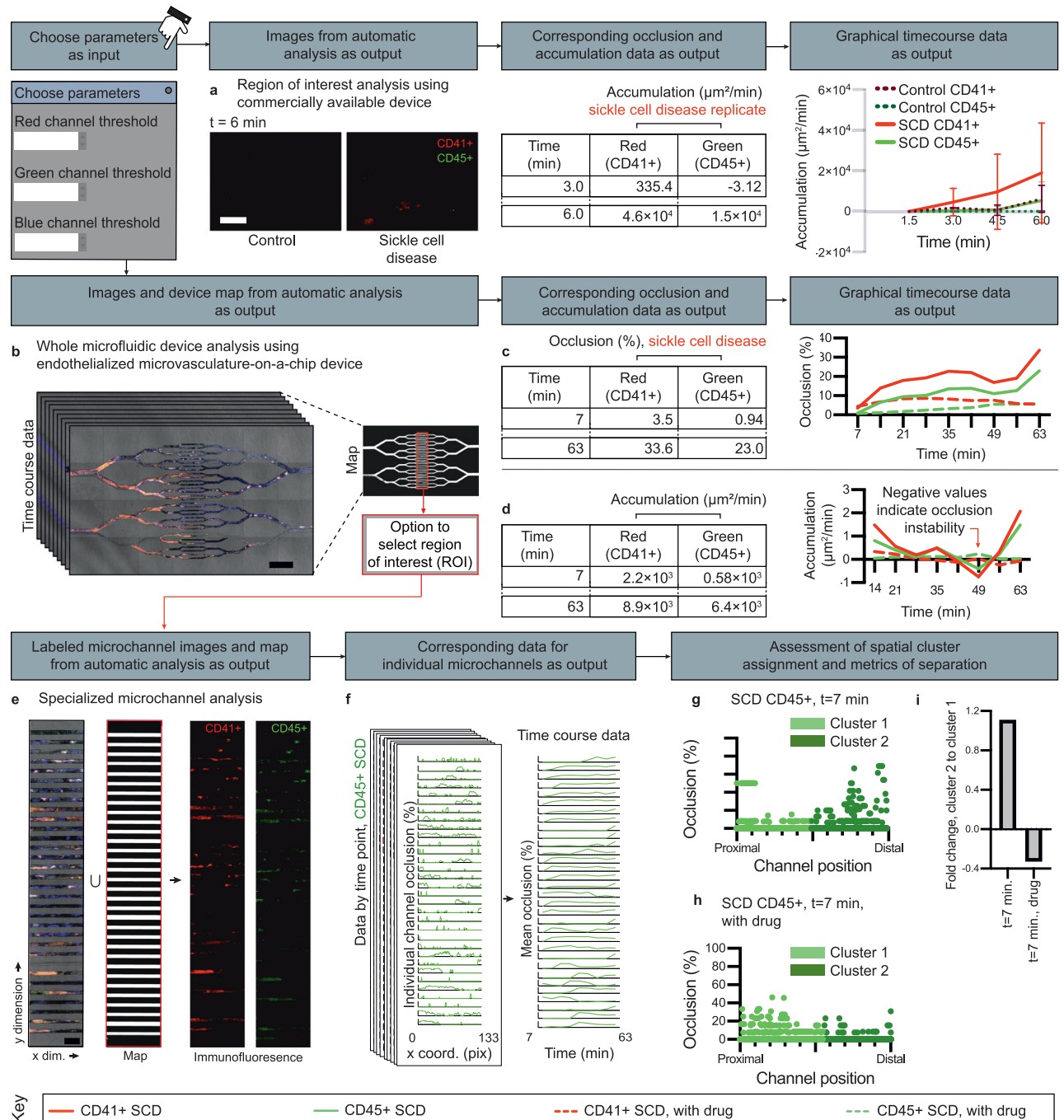

**Fig. 6 | iCLOTS multiscale microfluidic cell accumulation applications characterize cell aggregation in a variety of experimental devices, including those that are commercially available.** Designed for use with fluorescence microscopy images such that multiple components of a cell suspension can be simultaneously monitored, users adjust threshold values for any image color channel(s) where immunofluorescence signal is present. **a** Line graphs representing occlusion and accumulation over time are automatically generated. Here, CD41+ platelets and CD45+ white blood cells from sickle cell disease patient whole blood accumulate on an ibidi chamber device coated with collagen at a faster rate than healthy control whole blood ($n = 3$ replicates). Data taken at ×20 magnification, scale bar represents 50 μm. **b** The application is designed to automatically generate a map of all signals present, e.g., the dimensions of a microfluidic device, shown here with a microvasculature-on-a-chip device designed to investigate the effect of crizanlizumab on sickle cell disease whole blood samples ($n = 1$ experiment). Data taken at ×20, scale bar represents 200 μm. Percent occlusion (**c**) and accumulation rate (**d**) changes over the course of an experiment, showing microvascular occlusion

instability. **e** A microchannel-specific application is available for spatial analysis of one or many straight microchannel portions of a microfluidic device. Data taken at ×20, scale bar represents 50 μm. **f** Spatial quantification of microvascular occlusion is automatically performed by calculating an occlusion percentage for each pixel point along the length of each microchannel. ML algorithms enable further analysis in microchannels by treating each x-coordinate and corresponding occlusion measurement from each channel as a data point. At the initial time course timepoint, $t = 7$ min, CD45+ white blood cells in SCD whole blood (**g**) and CD45+ white blood cells in SCD whole blood treated with drug crizanlizumab (**h**) occlude endothelialized microchannels to variable degrees at each point along the 32 analyzed microchannels. **i** CD45+ white blood cells in SCD whole blood predominantly occlude distal ends of microchannels at early timepoints, while CD45+ WBCs in SCD whole blood treated with crizanlizumab occlude proximal entry points of microchannels at early timepoints. Source data are provided as a Source data file.

**Table 1 | iCLOTS comprises multiple workflows designed to address the needs of the widest range of experimental microscopy assays, with a focus on microfluidics**

| Workflow name | Adapted algorithm | Experimental device | Input data | Numerical result | Graphical result |
|---|---|---|---|---|---|
| Single cell tracking applications (Fig. 3a–d, Supp. Figs. 3, 4) | | | | | |
| Single cell tracking, option for channel flow or deformability | Crocker and Grier particle detection and linking | Commercial or custom microfluidics | Video Microscopy | Single-cell size and velocity with optional fluorescence intensity | Single-metric histograms, pairwise metric scatter plots |
| Cell suspension tracking (Fig. 4a–e, Supp. Fig. 5) | | | | | |
| Velocity time course and profile | Shi-Tomasi corner detection and Kanade-Lucas-Tomasi feature tracking | Commercial or custom microfluidics | Video microscopy | Time course velocity, velocity profile | Scatter plot indicating mean, minimum, and maximum velocity values per frame and line graph showing channel-wise profile |
| Adhesion applications (Fig. 5, Supp. Figs. 6–9) | | | | | |
| Cell morphology (Fig. 5a, b) | Crocker and Grier particle detection | Static slide, commercial microfluidic, custom microfluidic, flow chamber | Single image(s) | Single-cell size and morphology, density | Single-metric histograms, pairwise metric scatter plots |
| Cell functionality (Fig. 5c, f) | Region analysis | | | Addl. fluorescence intensity | |
| Single-cell protrusion characterization (Fig. 5d) | Harris corner detection | | | Addl. protrusion count | |
| Transient adhesion time (Fig. 5e) | Crocker and Grier particle linking | | Video microscopy | Addl. adhesion time | |
| Multiscale microfluidic accumulation applications (Fig. 6, Supp. Figs. 10, 11) | | | | | |
| Surface (Fig. 6a) | Region analysis | Commercial or custom microfluidics | Single or time course image(s) | Multi-color channel occlusion and accumulation | Time course line graph of accumulation, occlusion over time |
| Complex geometry (Fig. 6b–d) | | | | | |
| Microchannel (Fig. 6e, f) | | | | Addl. values for multiple regions | Addl. representations of multiple regions |
| Machine learning toolkit | | | | | |
| Clustering (Fig. 3e, f, Fig. 5g, h, Fig. 6g–i, Supp. Fig. 2) | K-means clustering | | iCLOTS or other numerical outputs | Cluster labels for individual data points, frequency distribution per sample, goodness of clustering statistics | Correlation matrix, scree plot, scatter plot with clusters indicated, mosaic plot |

interpretation process, the authors have prepared extensive application-specific help documentation, available within the software and at https://www.iCLOTS.org/.

In summary, we present iCLOTS, an interactive, freely-available software that addresses the clear need for widely-accessible, automated, and adaptable analytical tools for microfluidic-centric microscopy data. Designed to address the unique set of requirements presented by the heterogenous, dynamic nature of cells under flow, this implementation of well-validated image processing capabilities transforms microscopy data into quantitative, high-dimensional resolution datasets which may be further assessed by ML algorithms. No longer limited by time-consuming and error-prone manual analysis, we report key scientific and clinically-relevant findings made possible only through high-throughput, detailed quantitative analysis. Using iCLOTS, we observe (1) stiff subpopulations of RBCs in SCD, showing potential for subpopulation quantification within larger samples, (2) blunted blood flow profiles in sepsis, suggesting a role for new clinical biomarkers, (3) changes in cell behavior in HPS that could form the basis for a new diagnostic, and (4) changes in SCD blood accumulation in microfluidics that provide mechanistic insight into new pharmacological therapies.

iCLOTS is shared with the greater research community with dedicated channels for feedback such that over time, as researchers contribute their own needs and workflows, the software will continue to develop and improve. With iCLOTS, the field-wide effort to develop innovative microfluidic assays is met with commensurate analysis tools accessible to researchers who need it most, thereby enabling the generation of novel hypotheses in biomedical science and clinical medicine.

## Methods
### Ethical statement
Consent for all healthy human blood samples was obtained according to Georgia Institute of Technology IRB H15258. All SCD patient whole blood samples used for cell deformability experiments, all HPS patient whole blood samples used for adhesion experiments, and all SCD patient whole blood samples used for accumulation and occlusion experiments were drawn after informed consent was obtained in accordance with a corresponding Emory University IRB protocol. SCD whole blood sample for the rheological measurement was collected at Children's Minnesota Hospital in Minneapolis, MN under approved protocol by the Institutional Review Boards at the University of Minnesota and Children's Minnesota.

### Computational methods
All computational methods and software features were written in Python version 3.7 (python, https://www.python.org/). All Python packages used to implement image processing and machine learning methods are freely available open source projects and are described within individual assay methods. Manual analysis to assess accuracy of iCLOTS and determine comparative analysis times was performed with measurement tools available in Fiji distribution of ImageJ version 2.1.0 (ImageJ, https://imagej.net/software/fiji/)[80].

### Microfluidic device preparation for all assays
Soft lithography molds were created using SU-8 series photoresists (Microchem), adhering to all manufacturer's suggestions. Polydimethylsiloxane (PDMS; Ellsworth Adhesives) was poured into molds and allowed to cure at 60 °C for at least 2 h. The PDMS devices were removed from the mold and device-appropriate inlet and outlet ports were created by punching holes ranging from 0.75 to 1.5 mm through the inlet and outlet channels of the device. PDMS was then bonded to #1.5 coverslips (Fisher), or to thin cured PDMS for endothelialized experiments, described below, using a plasma cleaner (Harrick Plasma).

### Single-cell tracking workflows
**Deformability assay microfluidic device preparation.** The cell deformability microfluidic device consists of a series of branching microfluidic channels surrounded by two large bypass channels designed to reduce potential for changes in pressure caused by microchannel clogging via slow-moving cells. Therefore, a range of cell velocities is indicative of a range of size and deformability phenotypes alone. Microchannel height was chosen such that cells must necessarily deform to transit the device.

**Brightfield red blood cell deformability assay.** RBCs from SCD patients, IDA patients, or healthy time-matched controls were isolated from whole blood and diluted in PBS, then perfused into the cell deformability microfluidic devices using a syringe pump (Harvard Apparatus). Videomicroscopy was acquired at a rate of 25 FPS (20x, Nikon Eclipse TE2000-U).

**Brightfield white cell line deformability assay.** Acute T-cell lymphoblastic (Jurkat) cell lines and acute promyelocytic leukemia (HL-60) cell lines were suspended in PBS and perfused into the cell deformability microfluidic devices using a syringe pump (Harvard Apparatus). Videomicroscopy was acquired at a rate of 25 FPS (20x, Nikon Eclipse TE2000-U).

**Fluorescent CD71+ reticulocyte deformability assay.** Reticulocytes were isolated using a series of gradient methods[81] and resuspended in PBS. A 1:100 ratio of CD71+ anti-human antibody (Miltenyi Biotec) was added and the sample was incubated for 15 min at room temperature. A 1:500 ratio of Alexa Fluor-568 goat anti-mouse secondary antibody (Invitrogen) was then added and sample was perfused into the cell deformability microfluidic devices using a syringe pump (Harvard Apparatus). Videomicroscopy was acquired at a rate of 25 FPS (20x, Nikon Eclipse TE2000-U).

**Single-cell tracking computational analysis methods.** Image background containing channel walls was removed using background subtraction algorithms implemented using Python package OpenCV version 4.5.3 (OpenCV, https://opencv.org/)[82] Python package Trackpy version 0.5.0 (trackpy, http://soft-matter.github.io/trackpy/v0.5.0/)[21] algorithms are used to detect and track cells. Trackpy detects particles represented by small image regions with a 2-D Gaussian-like distribution of pixel brightness. Particle-linking methods connect cells into trajectories from which velocity values are calculated. Software users interactively choose parameters specific to their microscopy data including maximum cell diameter and minimum total pixel intensity to reduce contributions of cell clusters and noise, respectively. iCLOTS retains the highest-quality data points by imposing requirements for the number of frames a cell is detected and a minimum total observed distance traveled. Single-cell velocity is calculated using cell displacement and the rate of imaging. A specialized x-direction channel flow single-cell tracking application, which was used for deformability velocity measurements, is also offered. Average fluorescence intensity of individual cells was calculated by taking the mean sum of pixel intensity of the indicated cell region from all frames the cell was detected.

### Cell suspension velocity workflows
**Deoxygenation cell suspension velocity assay.** RBCs were isolated and suspended in Dulbecco's phosphate-buffered saline (DPBS) at a final hematocrit of ~25%. The resuspended RBC sample was subsequently loaded into a 3-layered microfluidic device designed for precise control of oxygen tension[30,31]. A pressure controller (Alicat Scientific) maintained a constant driving pressure through the device during video acquisition. Videos were recorded at ×40 magnification and 296 FPS on a Zeiss Axio Vert.A1 (Carl Zeiss) with a FLIR

Grsshopper3 camera (FLIR Systems) using the Image Acquisition Toolbox in Matlab (MathWorks). During video recording, the gas supplied to the device was switched from super-physiologic oxygen levels (21% $O_2$, 5% $CO_2$, 74% $N_2$) to anoxic conditions (95% $N_2$, 5% $CO_2$) to induce HbS polymerization.

**Cell suspension in sepsis velocity assay.** A microfluidic device with a straight channel portion (10 μm tall, 70 μm wide) was incubated with 1% (g/mL) BSA in PBS to coat the exterior and minimize interactions between the device wall and blood components. The presented microfluidic dimensions were chosen to approximate sheet flow for investigation of cell aggregation, but underlying computational methods also work for iso-symmetric flow[83]. The samples were then perfused using a syringe pump (Harvard Apparatus) under flow-controlled conditions to achieve target shear rates of 350 $s^{-1}$ and 175 $s^{-1}$ based on an analytical solution to laminar flow in a microfluidic channel.

**KLT velocity tracking computational analysis methods.** OpenCV was used to detect image features based on spatial intensity differences using Shi-Tomasi corner detection[23]. Movement between detected image features was calculated using OpenCV implementation of Kanade-Lucas-Tomasi optical flow[24]. Velocity of each feature is calculated using cell displacement and the rate of imaging. Profiles are generated by binning individual feature velocities according to their initial distance from the channel center using a user-set bin width.

## Cell adhesion workflows

**Brightfield microscopy platelet adhesion assay.** Isolated platelets were diluted in Tyrode's buffer. Platelet suspension concentration was optimized to ensure the measurement of single platelets and not platelet aggregates. Platelet suspension was incubated on coverslips (Fisher) that had been previously incubated with 100 μg/mL of human fibrinogen (Enzyme Research Laboratory) or Type 1 rat tail collagen (VWR)[25]. Platelets were allowed to adhere for 2 h at room temperature.

**Brightfield microscopy red blood cell adhesion assay.** Microfluidic devices with four separate channels (46 μm tall, 100 μm wide, 4 mm long) were coated with laminin derived from human placenta (Sigma) for 2 h at room temperature[58]. RBCs were isolated and resuspended in PBS to 0.2% hematocrit. Devices were perfused with RBC suspension via syringe pump (Harvard Apparatus). Images were acquired using a Keyence BZ-X810 microscope with a 20x/0.8 objective.

**Brightfield microscopy adhesion computational analysis methods.** Individual cells are located as particles using Trackpy as described for previous iCLOTS methods. Calculated values for eccentricity and radius of gyration of each cell particle are reported as circularity and radius, respectively. Radius of gyration is approximately equal to radius for red blood cells and platelets in brightfield microscopy images.

**Neutrophil isolation for all assays.** Neutrophils were isolated from whole blood collected in EDTA using a whole blood human neutrophil isolation kit (Miltenyi MACSxpress) and resuspended in PBS.

**Transient neutrophil adhesion assay.** 0.1 M N-Formylmethionyl-leucyl-phenylalanine (fMLP; Sigma) was added to neutrophil solution, which was then perfused through a straight channel microfluidic device (38 μm tall, 100 μm wide). Videomicroscopy was acquired at a rate of 25 FPS (10x, Nikon Eclipse TE2000-U).

**Transient adhesion cell tracking computational analysis methods.** Individual cells within each image are located as particles as described for previous iCLOTS computational analysis methods. A Trackpy

linking algorithm connects individual particles detected across different frames into pathways representing movement across the length of the device[21]. Transit time is calculated using cell displacement and the rate of imaging.

**Fluorescence microscopy neutrophil adhesion assay.** Isolated neutrophils were suspended in PBS and incubated with 10 μg/mL CellMask Deep Red plasma membrane stain (Invitrogen) and 5 μM SYTO 13 Green nucleic acid stain (Invitrogen) for 15 min at 37 °C, then fixed with paraformaldehyde (Sigma) for 10 min at room temperature. The cells were added to a non-coated glass chamber. Image data was acquired using Zeiss LSM 700 with a 40x/1.4NA Plan Apochromat objective.

**Fluorescence microscopy platelet adhesion assay.** Platelets were diluted to ensure the measurement of single platelets and not platelet aggregates and were incubated on coverslips prepared with the coating specified as the experimental condition[84]. Platelets were imaged on either Zeiss LSM 780/ELYRA PS1 confocal microscope using a 100x/1.46NA Plan Apochromat lens or an Eclipse Ti2 inverted microscope using a 40x/1.30NA Plan Fluor lens.

**Fluorescence microscopy adhesion computational analysis methods.** Selected red, green, and blue (RGB) image channels are binarized using OpenCV with a threshold value provided by the software user for each channel as an interactive input. Python package scikit-image version 0.18.3 (scikit-image, https://scikit-image.org/)[15] is used to calculate characteristics of each region that represents an individual cell within the binary image, including location of region centroid, area, and circularity. Summed functional stain fluorescence pixel intensity of the region above the threshold is determined using indexing methods and texture is calculated as the standard deviation of pixel intensity of the region. Individual regions of intensity, such as nuclei lobes or RNA SPOT signal, are calculated as local maxima of color channel intensity within the cell region using scikit-image.

**Fluorescence microscopy adhesion protrusion-counting computational analysis methods.** Images are binarized using a threshold value provided by the software user and regions representing cells are identified as described for previous iCLOTS computational analysis methods. OpenCV corner detection algorithms based upon the work of Harris and Stephens[22] are used to detect sharp points along the circumference of the convex region that represent filopodia-like protrusions using sharpness and minimum protrusion separation distance parameter values provided by the software user. iCLOTS reports minimum, mean, and maximum length of protrusion ends from the centroid of the region.

## Multiscale microfluidic accumulation workflows

**Commercial microfluidic device occlusion and accumulation assay.** Ibidi 0.2 mm μ-Slide I Luer with ibitreat devices were coated with collagen IV (VWR) and. In order to facilitate perfusion, syringes were loaded with whole blood treated with 40 μg/ml corn trypsin inhibitor (CTI; Haematologic Technologies), Anti-CD45 Mouse Monoclonal Antibody (VWR) at a ratio of 1:100, Integrin alpha 2b/CD41 Antibody (VWR) at a ratio of 1:200, and 6 mM $CaCl_2$. Blood was perfused using constant flow via a syringe pump (Harvard Apparatus) into devices for a period 10 min. A Keyence BZ-X810 Fluorescence Microscope and a 10× 0.8 NA lens was used to take a tile scan spanning the width of the microfluidic device was captured every 1.5 min.

**Microvasculature-on-a-chip device occlusion and accumulation assay.** Endothelialized branching microfluidic devices with 32 microchannels (30 μm wide, 30 μm tall, 200 μm long) were prepared by growing a confluent layer of human umbilical vein endothelial cells (HUVECs, Lonza, cat. # cc-2519) on the surfaces of the channels of the

microfluidic device[73]. Once devices reached endothelial cell confluency, CellMask Deep Red Plasma Membrane Stain (Fisher) was added to the culture media. In order to facilitate perfusion, syringes were loaded with whole blood treated with 40 μg/ml corn trypsin inhibitor (CTI; Haematologic Technologies), Anti-CD45 Mouse Monoclonal Antibody (VWR) at a ratio of 1:100, Integrin alpha 2b/CD41 Antibody (VWR) at a ratio of 1:200, and 6 mM $CaCl_2$. Blood was perfused using constant flow via a syringe pump (PhD Ultra, Harvard Apparatus) into devices for 28 min. Using a Zeiss LSM 700-405 confocal microscope and a 10x/0.8NA lens, a tile scan of the entire microfluidic device was captured every 7 min.

**Occlusion and accumulation computational analysis methods.** Selected RGB image channels are binarized using Open-CV with threshold values provided by the software user for each color channel as an interactive input. To generate a map of the experimental device channels, each RGB image with threshold applied is summed into a one layer image and an additional set threshold is applied. In microchannel experiments, a left-right indexing operation is performed on each individual channel as detected by scikit-image to smooth coordinates of channel walls. Percent occlusion of a full device or an individual microchannel was calculated from ratio of area of the device with signal to the total area of the device. Accumulation from sequential images is calculated as the change in signal area over time.

### Validating computational methods
iCLOTS has been designed to produce accurate, robust quantitative results in a fraction of the time required for manual analyses. To validate iCLOTS, when possible, we have performed a statistical comparison of iCLOTS application results to manual analysis performed by one or several expert hematologists (Supplementary Figs. 3, 7 and 8). For all applications, we have performed sensitivity analysis (Supplementary Figs. 3, 5, 7 and 8). Sensitivity analysis is a type of computational analysis where an algorithm is applied to the same dataset multiple times, each time with different parameter values. In the context of the iCLOTS software, parameters are numerical factor(s) that set the conditions of the algorithm's operation. Sensitivity analysis is designed to show that algorithm results are robust, i.e., reasonable changes in parameters do not result in substantially different interpretation of results.

### Reporting of numerical and graphical results
Python package pandas version 1.3.3 (NumFOCUS, https://pandas.pydata.org/)[18] was used to prepare and export numerical data as an excel file. Large datasets may be exported as a comma-separated value (CSV) file. Python package matplotlib version 3.4.3 (Matplotlib, https://matplotlib.org/)[19] was used to prepare and save graphical data. Python package Seaborn version 0.11.2 (seaborn, https://seaborn.pydata.org/index.html) was used to prepare specialized pairplot data[20]. Pairplots are a specialized set of graphs where each graph compares two variables using a scatter plot labeled with condition. Graphs on the diagonal of the pairplot are histograms of a single metric.

### Machine learning interpretation
All applied machine learning algorithms were implemented using scikit-learn version 1.0.2 (scikit, https://scikit-learn.org/)[16]. iCLOTS presents tools to implement K-means clustering algorithms[41]. The software user has the option to upload one or several iCLOTS-generated excel output files, each of which comprises a sample label. The software user then chooses features, or numerical descriptors of individual cells or events, from columns within the excel document. All data points from all sample labels are combined, and a scree plot is generated to provide a suggested optimal number of mathematically defined clusters[43]. All data points are clustered into a software user-provided $n_{clusters}$ and all sample label data points are returned with a

corresponding cluster label for further statistical interpretation. Statistics including silhouette score[44] are returned to assess relative goodness of clustering, with values approaching 1 indicating most distinct cell populations (Supplementary Fig. 2).

### Statistics and reproducibility
The primary scientific value and purpose of our manuscript is to introduce a tool to a series of users by introducing each application and providing an experimental case study that demonstrates inputs, parameters, outputs, and potential use cases. The major innovation of this paper is a standalone, free-to-use, interactive image analysis and machine learning software. As such, our case studies in this manuscript are primarily designed to give readers a sense of how the software can enable their experiments. Experimental data was excluded only when imaging quality was not sufficient for computational analysis. Authors responsible for computational analysis were blinded to experimental sample classification, if any. No statistical method was used to predetermine sample sizes.

### Reporting summary
Further information on research design is available in the Nature Portfolio Reporting Summary linked to this article.

## Data availability
The imaging data generated for this study sufficient to recreate the results of this manuscript and demonstrate all software functionality to users is included as test data at https://www.iCLOTS.org/, however, the full imaging dataset is available within five working days, without restriction, from the corresponding author Wilbur A. Lam upon request. All numerical source data generated in this study are provided in the Source data file. Source data are provided with this paper.

## Code availability
All current and past versions of standalone software are available at https://www.iCLOTS.org/software. All software source code is available at https://www.github.com/iCLOTS. All methods as standalone scripts are available at https://www.github.com/LamLabEmory.

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

## Acknowledgements

The authors thank Cassie Mitchell, Yumiko Sakurai, Ethan Castellino, and Andrew Shaw (Parker H. Petit Institute for Bioengineering and Bioscience, Georgia Institute of Technology) for technical assistance. This work was supported by National Institutes of Health, Institute of Heart, Lung, and Blood grants R01HL130918, R01HL140589, and R35HL145000 (W.A.L.), 5T32HL139443-03 (J.O.M.), HL160210-01 (O.O.), T32HL139443-3 (S.S.A.), and L40HL149069 (C.C.), and National Institute of Allergy and Infectious diseases grant R38AI140299 (E.I.). This work was performed in part at the Georgia Tech Institute for Electronics and Nanotechnology, a member of the National Nanotechnology Coordinated Infrastructure, which is supported by National Science Foundation, division of Electrical, Communications, and Cyber Systems grant 1542174.

## Author contributions

C.M.B. provided clinical blood samples. O.O., E.I., K.S.F., J.O.M., C.C., S.S.A., H.C., S.H., K.W., E.N.E., and C.K.K. performed experiments. M.A.C., M.L.K., V.A.S., and D.K.W. provided experimental and data analysis guidance. E.I., S.H., D.Y.Z., and J.M.V. contributed to computational methods. M.E.F. performed data analysis and developed software. R.G.M. and M.E.F. prepared software for distribution. M.E.F. and W.A.L. wrote the paper, and all authors contributed in editing the paper.

## Competing interests
The authors declare no competing interests.
