## [Peer Review File · Nature Communications]

REVIEWER COMMENTS

Reviewer #1 (Remarks to the Author):

In this manuscript Fay and colleagues describe the iCLOTS software toolbox they have developed for quantitative analysis of a multitude of static and dynamic experimental approaches that are used to describe hemostatic processes. While many of these analyses are already possible using common scientific imaging analysis programs such as ImageJ, such methods are often not standardized, depend on specialized plugins or writing of dedicated macros and require significant user intervention before meaningful data can be extracted. Specialized software packages such as iCLOTS that combine many common hemostatic analyses in a user-friendly environment are therefore a welcome contribution to our field.

The purpose of this manuscript is to showcase the capabilities of iCLOTS by analyzing a number of datasets of adhesion of cells or proteins to adherent surfaces or microfluidic channels. Robustness of their analyses is demonstrated using sensitivity analysis and comparisons with manual quantification methods. The paper is well written and easy to follow, the methods are described in sufficient detail.

I only have a number of minor comments that mostly relate to usability and interface of the iCLOTS software.

To get a feeling for its functionality I have tried to use the iCLOTS program using some of the test data sets that were supplied by the authors.

I found that the user interface does not scale well with the window size of the iCLOTS program. When images or videos are displayed, they remain fixed in the same small size even when stretching the window to cover the entire screen. At such limited image size, some of the object detections are difficult to discern, for instance the platelets adhered to collagen/fibrinogen matrix, and assessing whether changes in input parameters enhance or decrease object detection becomes difficult.

On page 4 the authors state: "The original image and the image as analyzed displays in the center of the window, with changes in inputs of parameters updating in real time, allowing users to fit algorithms to their specific set of data (Figure 2D)."

On my setup (PC with Windows 10) the iCLOTS program did not update detections setting in real time. For instance, for the detection of platelets adhered to collagen or fibrinogen, changes in the "Maximum diameter (pix)" or in the "Minimum intensity (a.u.)" parameters did not automatically update the object detection, but only when first clicking "Set detection parameters" and then select "Dark cells on light background". Or in case I loaded multiple images at the same time, object detection was updated after flicking from one image to the other and back. This makes setting the right input parameters a tedious process. Same for object detection in videos, this only updated when going one frame forward (and then back), or when setting detection parameters and selecting dark/light cells on light/dark background.

In video analyses (for example the transient adhesion time analysis) I found it very difficult to scroll through the frames. Only a very small sliding bar is present with which one can scroll through the frames but this is very insensitive and goes in steps of 6 frames. Please add buttons or some sort of explanation how to scroll through the video frame by frame.

Page 8, paragraph describing case 3: Figure 4F should be Figure 5F I presume?

Neutrophil adhesion (Figure 5E) is quantified in numbers of neutrophils adhered and their transient adhesion times. Is it possible to analyze neutrophil crawling (motility and/or directionality, extravasation) using iCLOTS? This would leverage iCLOTS for applications that study transendothelial migration of various immune cells.

Reviewer #2 (Remarks to the Author):

The authors present a software suite for analyzing time-dependent cell data, especially in microfluidics devices and blood related studies. Currently, most dynamic cell studies require custom code development and manual steps during analysis, and the authors' software, iCLOTS, aims to provide a simple to use solutions package. Overall, the concept is similar to popular open source software such as ImageJ and FIJI, but with emphasis on time dependent microscopy data that are common in blood related studies. There is strong merit to having such software and it can benefit many users. Some comments to consider:

1) Integrating iCLOTS as part of ImageJ can leverage the many existing plugins available for ImageJ. This would seem to improve data analysis capacity for end users and is consistent with the central goals stated by the authors.

2) While this software can be very helpful especially to those with less analytics experience, this study in its current form appears to be primarily generating a collection of existing tools along with tailoring them toward specific applications in analyzing blood. The demonstration of specific applications in the case studies is intriguing and can significantly raise the impact of this study. However, currently there is limited depth to the case studies and the patient sampling is low. The impact would increase if novel insights can be generated and validated, e.g. validation of specific biomarkers rather than suggestive findings, which will require larger sampling and patient numbers and addressing potential confounding factors. As presented, findings are only suggestive of potential applications, but without validation of new biomarkers or discovery of novel mechanistic insights. While this may require significant effort, it will raise the impact and novelty of the paper, especially if a new biomarker is validated and can be immediately used clinically or if a new mechanistic insight about a disease mechanism is discovered.

3) The authors emphasize the need for simple to use quantitative tools for characterizing cell features in time-dependent and microfluidic experiments and how their software addresses this need. It would be useful if the authors take existing data from the literature, especially from other lab groups and other experimental designs, to demonstrate effectiveness of their software in extracting useful information, including replicating some results of those studies. This will help demonstrate ease of use, amount of manual customization required or to be expected from end users, and applicability for other device designs and biomedical applications beyond the case studies here. The authors can also discuss if there are certain experimental designs that are required in order for the software to be immediately usable and the potential limitations.

4) The filopodia tool is counting something similar to corners, but those may not be actual filopodia which are driven by certain molecular factors such as myosin X and fascin. A test that demonstrates the percentage of false positives/false negatives can be helpful.

Reviewer #3 (Remarks to the Author):

On a general level, the main and unique result of this work is an overall software platform that integrates imaging, data analysis, feature extraction, machine-learning and interpretation tools through a guided step-by-step strategy. Inputs can be obtained from different microscopy or microfluidic sources, making the whole system very flexible and potentially useful for many users.

In general, the whole work is excellent and has high potential impact.

As for the users, they are expected to be biomedical researchers. In the paper, the authors frequently mention the role of the user in configuring different parameters throughout the workflows and tend to state that the system is easy to use. This implies knowledge in several disciplines. I would recommend the authors to discuss whether such a system could also be accessible to clinical laboratories (also mentioned in the paper), what could broaden the scope of practical use of their system.

The work is truly original and robust and can have a scientific and practical impact. Compared to previous systems, iCLOTS is automated versus manual techniques, and is independent of imaging

sources. It is also free and open access, compared to commercial systems.

The work has presented four applications involving blood cells where analytical needs are highly relevant. All cases have been satisfactorily addressed and are describe in full detail. The conclusions are supported by these results and the discussions delivered.

There are different techniques involved in this work and applied to four relevant cases. Not apparent flaws have been detected in the data analysis and interpretations appear correct.

The different methodologies implied in this work seems appropriate within the state of the art. The main innovation lies in the overall strategy and how the different tools are nicely combined to build the system.

I have two remarks that authors should discuss more precisely. They refer to the image quality. Authors mention a robust design around data sets from multi groups. The concept of robustness should be more clearly defined and the measures taken to make a robust design should be explained. In the same context, the dependence on the image quality is associated to "label all analyzed imaging data with single-cell indices corresponding to numerical data, allowing users to assess potential outliers". This sentence is found unclear and unrelated to image quality.

The core paper and the supplemental material are well written and provide enough detail on the methods. Qualified researchers could reasonably reproduce the results. The authors also offer public documentation on a web site.

Reviewer #4 (Remarks to the Author):

The authors have developed a suite of open source artificial intelligence-enabled software for analyses of blood cells in microfluidic and microscopy-based assays. They demonstrate the capability of their software by passing clinically relevant blood samples (whole blood or washed cells) through microfluidic devices mimicking the microvasculature. Analysis is performed on both capillary and venular sized rectangular channels.

The focus of the manuscript is to provide other researchers with access to advanced analysis tools for microfluidic studies of blood and other cell types.

Although I strongly support the goals of the authors I have a number of concerns. For wide use by researchers who are not experts in this type of analysis there needs to more guidance on the practical use of the software and better integration of the main text with the Supplement. Practical advice is missing for why specific choices were made for the microfluidic experiments. Why would one chose a fixed flowrate with a syringe pump vs constant pressure perfusion? What range of hematocrit levels to use in the feed reservoir for RBC suspensions and why? When do you use bright field illumination vs fluorescence and why? Why were all of the experiments run at room temperature when temperature has a significant effect on RBC deformability and cell physiology?

For many of the examples given, there are other methods for obtaining a matching dataset to verify the accuracy of the analysis algorithms and interpretation of the microfluidic results. It would strengthen the paper to include comparison to "gold standard" measurements. For example: an advantage of the microfluidic approach is direct comparison between cell velocity and cell size but how does the cell size measurements compare with other measures of cell size or volume? The one case given of manual single cell RBC velocity measurements does not fully address the validity of the velocity algorithm (see below).

The manuscript also lacks details on accuracy of the various algorithms and spatial resolution of the optical flow method. Although there is a sensitivity analysis for some algorithm parameters the paper lacks data on reproducibility, i.e. analysis of repeated runs of the same sample. I would also expect a critical evaluation of the limitations of the different analysis techniques as well as the

microfluidic approaches presented. Some indication of the minimal computer system requirements would be helpful.

Specific Comments

1. Figure 3 shows an example of single-cell tracking applications which provide high-resolution measurements of velocity relative to cell size and fluorescence intensity. The figure demonstrates “use of this application with a specialized microfluidic assay where velocity of a cell transiting a microchannel indicates relative cell stiffness.”

a. I assume the microfluidic device referred to is the one shown in Figure 1 Panel D, and the device was perfused at a constant rate using a syringe pump as described in the Supplement (there needs to be better integration between the main text and the supplement)

b. Why is there such a wide distribution of RBC velocities as shown in Panel B and C with a constant perfusion rate, a symmetric branching pattern for the microfluidic channels and what appears to be a very low RBC density in the suspension? I would expect a narrow range of velocities with the distribution skewed towards low velocities (channels with high velocities should have a higher RBC flux and hence there should be a higher frequency at high velocities). Is the wide distribution of velocities for control RBCs due to an artifact introduced by the syringe pump or variability in dimensions of the microfluidic channels causing asymmetric distribution of flow? Or is it due to artifacts introduced by the analysis software?

c. Why did you chose to show in the Table in Panel B an iron deficient RBC with half the pixel size and half the velocity of a control RBC? This reinforces the concept that iron deficient RBCs are both smaller and stiffer but in the graph in the same Panel there doesn't appear to be a relationship between iron deficient RBC size and RBC velocity. Since these iron deficient cells are stiffer I was expecting a negative relationship. I was expecting that RBC velocity would increase as RBC size decreased; hence as flow resistance decreased. At least that is what I would expect with a constant driving pressure microfluidic system. But the authors have used a syringe pump to perfuse the device, i.e. constant flowrate system. With a constant flowrate, RBCs regardless of their size and stiffness should be forced through the channels at the same velocity. A constant perfusion system is not sensitive to cell deformability.

d. Supplement: Deformability assay microfluidic device preparation “Microchannel height was chosen such that cells must necessarily deform to transit the device. A microchannel height of 5 μm was prepared for RBCs and reticulocytes and a microchannel height of 13.8 μm was prepared for WBCs and cell lines.” But the width of microchannel was 5.9 μm which means the diagonal dimension for RBC and reticulocyte transit was 7.7 μm . Most normal size RBCs ($\sim 8.0 \mu\text{m}$ in diameter $\times 2.2 \mu\text{m}$ thick) would require minimal deformation to pass through these channels since they are not constrained to orient to the smallest dimension. Smaller iron deficient RBCs could pass without needing to deform at all. I am not convinced that the channels are not small enough to test RBC deformability based on their velocity. It also isn't clear why the channels needed to be twice the size for WBCs since they also traverse the same capillaries as RBCs in vivo.

e. With a constant flowrate generated by the syringe pump, why did the iron deficient RBCs appear to have a lower mean velocity?

2. In Supplement Fig 3, iCLOTS single cell tracking is compared to manual analysis by three expert hematologists. No statistical analysis is provided and only the shape of the velocity distribution is given as a visual comparison. The discrepancy between manual measurements and iCLOTS, and between the three experts appears to be more than simple measurement error of a few micrometers due to locating individual RBCs. I suspect the differences reflect the challenge of identifying a discrete RBC from one frame to the next when there are multiple nearly identical RBCs in the channel. This is especially challenging at velocities shown in the distribution where the RBC are traveling from 2 to 6 times their length in a single frame. The data should be available to do a Bland-Altman analysis to detect systematic errors between experts and between experts and the tracking algorithms.

<https://pubmed.ncbi.nlm.nih.gov/32456091/>

3. Septic Whole Blood

Loss of RBC deformability in sepsis and activation of coagulation pathway are well established in animal models of sepsis and septic patients. The ability to study altered blood flow in microfluidic channels reflecting conditions found in venules could be an important clinically relevant tool. However as presented I have a number of concerns.

a. In Figure 4 Panel A, testing sickle cells, the full width of the channel is shown (walls are visible although not labelled). In Panel B, the walls are not visible and although the channel images are same size the scale bars are different. It is not clear that the channels are 70 μm wide as described in the Supplement. Please clarify and redo images to show the walls. NOTE: velocity units in Panels A & B are incorrect. Please give explanation for the blue dots in the figure caption.

b. Channels perfused with "whole blood samples at shear rate of 350 sec^{-1} chosen to recapitulate venous shear rate of a vessel of similar dimensions." The reader is left with impression that this channel mimics a 70 μm diameter venule at a physiological relevant wall shear rate. But the channel is highly asymmetric with a depth (or height) of only 10 μm (Supplement: Cell suspension in sepsis velocity assay) which is much closer to sheet flow than to flow in a tubular vessel. Which wall was the shear rate calculated for? In vivo a 70 μm diameter venule has a much higher velocity than 500 $\mu\text{m/s}$ so I assume the flowrate was set to match the wall shear rate along the 10 μm dimension (top or bottom of the channel, not the sides). Please clarify.

c. Panels C & D in Figure 2 show that the velocity profile for the septic blood is both blunter and at a higher velocity than the control blood. As shown the septic blood appears to have a substantially higher flowrate. Please explain how this is possible if both types of blood samples are being delivered by syringe pump at the same flow rate. Was the video captured far enough downstream of the entrance of the channel to ensure a fully developed flow profile for both samples? Was the focal plane for the video capture set to the same depth into the channel, e.g 5 μm from the wall or to the depth with maximum velocity? Or is there a systematic error in the optical flow calculation related to the blood sample type (individual cells vs cell aggregates)?

d. What is the spatial resolution of the optical flow method across the channel for the window size used? Supplement Fig 4 Panel B shows velocity profiles for different window sizes. Regardless of the window size in the dimension across the channel (h), there is a velocity value at +35 and -35 μm (supposedly at the walls). Explain how the optical flow algorithm can calculate velocity at the wall regardless of the window size. Give the window size in μm as well as pixels.

e. Finally, there is usually a relatively cell free plasma layer at the wall of blood vessels of this size due to migration of RBCs from higher shear rate at wall to lower shear near centreline and due to RBC collision with the wall. In Fig 4 Panel B there is no evidence to suggest a pseudo cell free layer at the edges of the channel and some RBCs appear to be cut-off suggesting the channel is wider.

4. Microvasculature-on-a-chip device occlusion and accumulation assay

"Blood was perfused using syringe pumps into devices for 28 minutes." Why was a syringe pump used for the occlusion and accumulation assay rather than constant pressure? As the channels become occluded the resistance across the microfluidic device increases. Hence, the driving pressure across the channels increases as the syringe pump forces a constant flow through the device (same argument as above). This dynamically changes the conditions for adhesion and occlusion over the 28 minute sample. With multiple channels the flow distribution among channels will also change due to the heterogeneous distribution of accumulation (as shown in Fig 6). Provide an argument for why a constant flowrate system was used and include the limitations.

Reviewer #5 (Remarks to the Author):

This paper presents iCLOTS, a toolkit available for free download. iCLOTS includes four categories of applications: cell adhesion, single-cell tracking, velocity profile, and multiscale microfluidic-centric applications. iCLOTS is open source, easy to use and requires no coding expertise to implement. This allows iCLOTS to be used in different experiments. New observations are also reported in this paper benefiting from the efficiency of the iCLOTS analysis. The iCLOTS is novel and interesting, and their experimental evaluation is in general convincing. However, there are some minor revision for the authors:

1. In Case study 3, the authors found that differences in adhesion to collagen-coated surfaces, platelet morphology, and phosphatidylserine (PS) exposure in platelet samples from patients with an FLI-1 mutation and HPS as compared to healthy controls. But there is no corresponding 'Figure 4F' in Figure 4 provided by the authors. There is also some ambiguity in the author's expressions. Does 'Supplementary Figure 6' correspond to the missing 'Figure 4F' or is it a comparison diagram for 'Figure 4F'? Supplementary Figure 6 provides several figures. Which one belongs to Case study 3?

2. In the second paragraph of the iCLOTS workflow section, the authors propose that iCLOTS can analyze data quickly, with processing times ranging from a few seconds to a few minutes, depending on the size and number of files, but do not provide specific data proofs. In some of the images, the authors mark the time of analysis, but do not specify it. When working with very large datasets, can iCLOTS also limit the analysis time to a few minutes? Please provide the specific data quantities and the corresponding analysis time.

3. In the second paragraph of the Discussion section, the authors suggest that blood samples are the most useful test benchmark, but do not give a detailed explanation and experimental proof. Please add an experiment to clarify it.

RESPONSE TO REVIEWERS' COMMENTS

Dear Reviewers,

We thank you for inviting us to submit a major revision of our manuscript “iCLOTS: open-source, artificial intelligence-enabled software for analyses of blood cells in microfluidic and microscopy-based assays” for publication in *Nature Communications*. We would like to thank the the five Reviewers of *Nature Communications* for their constructive review, time, and thoughtful consideration of our original manuscript and we were very encouraged by your interest in our open-source software and associated experimental findings. **In response to the Reviewers' valuable feedback, we have addressed each and every concern raised**, which has undoubtedly improved the quality of our manuscript and made our software a more accessible and useful product for our target audience, the larger community of biomedical researchers and clinicians.

In addition to the many positive comments, including that our manuscript is “well written and easy to follow” (Reviewer 1), that “there is strong merit to having [this] software and it can benefit many users” (Reviewer 2), that the work is “excellent and has high potential impact” (Reviewer 3), that they “strongly support the goals of the authors” (Reviewer 4), and that the work is “novel and interesting” (Reviewer 5), we identified five primary concerns based on the feedback from Reviewers: (1) lack of validation of new biomarkers, (2) need for larger patient samples, (3) requests to replicate previous findings, (4) benchmarking results against clinical gold standards, and (5) implementation of pressure-driven flow/experimental assay concerns.

Below we address these five primary concerns and then address every concern of each reviewer individually. **These primary concerns followed by our responses and then the concerns raised by each individual Reviewer which again are followed by our responses are all indicated with blue bolded text.** All original reviewer comments are underlined. Responses are given in bulleted lists below the comment they directly address. “*Relevant text from the manuscript or supplement is in italics with revised changes in yellow highlight.*” **Key components of our response are bolded.** Please note that the iCLOTS project also incorporates online software “documentation,” an extensive user guide containing information about application methods, inputs, parameters, outputs, and experimental considerations, at <https://www.iCLOTS.org/documentation> and within the “tutorial” buttons available in each application of the standalone software. “*Relevant text from the online and in-software user guide/documentation is in purple italics with revised changes in cyan highlight.*” Our revisions in response to the Reviewers' feedback have led to the addition of one updated Figure, two updated Supplementary figures, six new Supplementary figures, one new Supplementary table, an additional **Software use** section within the Supplementary methods and online user guide/documentation, and a new section of the online documentation discussing practical experimental design concerns.

(1) Concerns with the lack of validation of new biomarkers

We thank the Reviewers for their feedback that the impact of our manuscript would increase if novel insights could be generated and validated, e.g. the validation of specific biomarkers rather than the suggested findings presented. Reviewer 2 has shared that if a new biomarker that could be used clinically or if a new mechanistic insight about a disease mechanism is discovered, it would raise the novelty of the paper.

- That said, our primary purpose of this manuscript is to introduce a new software tool to multiple scientific and researcher user groups, as other similar *Nature Communications* publications have done, including the recent manuscripts describing B-SOiD, an unsupervised algorithm for identification and fast prediction of animal behaviors (Hsu et al., *Nature Communications*, 2021) and BUNDLE, a tool for inferring localization of proteins (Crook et al., *Nature Communications*, 2022). We were also inspired by the manuscripts introducing the open-source software projects ilastik (Berg et al., *Nature Methods*, 2019) and CellProfiler (Carpenter et al., *Genome Biology*, 2006), which followed a similar approach of introducing a tool then providing a brief case study designed primarily to demonstrate utility. **Therefore,**

the major innovation of this paper is an interactive toolkit that addresses a need for a field stymied by a lack of analytical tools for innovative, physiologically-relevant assays of any design, with a special focus on methods incorporating fluid flow, dynamic adhesion, and/or commercially-available or custom-made microfluidics. To that end, we've created a piece of standalone, free software that democratizes use of well-validated image analysis and machine learning algorithms for all end-user biomedical researchers who would benefit from advanced computational methods.

- However, we wholeheartedly agree with the reviewers that validation of a key biomarker that provides novel mechanistic insight would add to the impact and novelty of our manuscript, and **in response to this suggestion, we have validated a key biomarker for SCD pathophysiology using our iCLOTS software** by collecting additional data that shows a reduction in CD41+ platelet and CD45+ WBC occlusion of *in vitro* microchannels after treatment with the newly FDA-approved drug crizanlizumab. **Our data validates that platelet and WBC occlusion in microfluidic channels can be used as an objective biomarker, enabled by iCLOTS, that correlates with clinical outcomes, specifically the efficacy of new sickle cell therapies like crizanlizumab.** Importantly, our new data is congruent with clinical trial results published in the *New England Journal of Medicine*, which led to the FDA approval of crizanlizumab for sickle cell disease. That paper demonstrates clinical efficacy of the drug while our data demonstrates that microfluidic occlusion using iCLOTS can serve as an objective biomarker for the clinical endpoints in that paper.
- Specifically, for this experiment, we have performed iCLOTS analysis using the multiscale microfluidic accumulation/occlusion application with four additional sickle cell disease patient samples (total n=5) with and without crizanlizumab treatment, **presented in a new Supplementary figure 11:**

“Supplementary figure 11. iCLOTS analyses results serve as effective biomarkers of pathophysiology congruent with findings from successful clinical trials. In experiments investigating the accumulation of CD41+ platelets and CD45+ white blood cells in a microvasculature-on-a-chip model (n=5 experiments), a ratio of microchannel occlusion without crizanlizumab treatment to microchannel occlusion with crizanlizumab treatment greater than 1 indicates that crizanlizumab is reducing the rate of microchannel occlusion within the microfluidic devices. Using the iCLOTS microfluidic

device-scale accumulation and occlusion application, we find that in n=5 sickle cell disease patient samples, the ratio of (A) CD41+ platelet occlusion and (B) CD45+ white blood cell occlusion for treatment without crizanlizumab to treatment with crizanlizumab indicates reduction in occlusion (ratio > 1) in 60% of samples. (C) Ataga et al. (New England Journal of Medicine, 2016) found a similar lowered percentage rate of crises per year (1.63 with high-dose crizanlizumab treatment as compared with 2.98 with placebo, indicating a statistically significant 45.3% reduction) and uncomplicated crises per year (1.08 with high-dose crizanlizumab treatment as compared with 2.91 with placebo, indicating a statistically significant 62.9% reduction.)”

- We have also referenced the new Supplementary figure 11 within the main manuscript text:

“The iCLOTS accumulation application automatically generates a map of complex microfluidic device dimensions (Figure 6B). We expectedly observed that both CD41+ platelets and CD45+ WBCs within whole SCD blood treated with crizanlizumab occlude microfluidic channels less than untreated SCD blood samples (Figure 6C) at a lowered percentage rate similar to completed clinical trials (Supplementary figure 11)⁶⁹.”

Where ⁶⁹ indicates the publication describing the referenced clinical trial:

Ataga, K. I. et al. Crizanlizumab for the Prevention of Pain Crises in Sickle Cell Disease. *New England Journal of Medicine* 376, 429-439, doi:10.1056/NEJMoa1611770 (2016).

- These new experiments indicate that using iCLOTS, **per the Reviewers’ suggestions, we can indeed validate new biomarkers, and in this case we show a reduction in CD41+ platelet and CD45+ WBC occlusion of microchannels after treatment with crizanlizumab** congruent with clinical trial findings published in the *New England Journal of Medicine*.

(2) A need to use larger patient samples

Several Reviewers shared concerns that there is limited depth to our case studies and patient sampling is low. We thank the reviewers for valuable feedback on a pathway to improving the impact and novelty of our paper through additional experiments. As such, we have added nine additional paired healthy/iron deficiency anemia patient samples (for a total of ten pairs) to our single cell tracking experiments, three additional paired healthy/SCD patient samples to our adhesion experiments, and four SCD patient samples (for a total of five samples) to our accumulation and occlusion experiments. The additional patient samples have resulted in new, statistically significant conclusions for each of these applications.

- That said, we believe the primary scientific value and purpose of our manuscript is to introduce a new tool to a series of users by introducing each application and providing an experimental case study that demonstrates inputs, parameters, outputs, and potential use cases. The major innovation of this paper is a standalone, free-to-use, interactive image analysis and machine learning software. **As such, our case studies in this manuscript are designed to give readers a sense of how the software can enable their experiments but are not meant to be scientifically conclusive. That is up to the user regarding how they want to use iCLOTS and how many patient samples they can accrue, but our software can make the process more efficient, less onerous, and generate more data.**
- However, **we do not disagree with the Reviewers that higher patient sampling would significantly raise the impact of this manuscript and therefore have added additional experiments with increased amounts of patient data** to the key findings from each additional category of image

processing applications (adhesion, single cell tracking, and multi-scale microfluidic accumulation/occlusion):

- Using the iCLOTS single cell tracking application, in response to the reviewer's request for additional sampling we have performed experiments with nine additional paired healthy control and patient samples to show that RBCs from iron deficiency anemia samples are significantly smaller and slower than healthy control samples in a specialized deformability assay (**n=9 additional paired samples, for a total of 10 paired samples**). Results are included in subfigures E and F in an updated Supplementary figure 3:

This updated Supplementary figure and caption show RBCs from iron deficiency anemia samples are significantly both smaller and stiffer than healthy control counterparts:

*“(E) An estimation plot, a method of presenting a t-test that displays raw data and the confidence interval for the difference between compared sample means, shows iron deficiency anemia RBC samples have a significantly lower mean cell velocity than their time-matched healthy control RBC samples (n=10 pairs, **p<0.01 via paired t test). (F) An estimation plot shows iron deficiency*

anemia RBC samples also have a significant lower mean cell size than their time-matched healthy control RBC samples (n=10 pairs, *p<0.001 via paired t test)."**

This additional experiment is also referenced in the main manuscript:

*"Here we show application adaptability using a range of blood cells obtained from clinical samples and cell lines, including red blood cells (RBCs) from patients with SCD (Figure 3A), RBCs from patients with iron deficiency anemia (Figure 3B, **Supplementary figure 3**), immortalized white cell lines (Figure 3C), and reticulocytes from patients with SCD (Figure 3D)."*

2. **Using our adhesion application, in response to the reviewer's request for additional sampling we have performed three additional experiments using patient samples that find that healthy control RBCs adhere to laminin-coated microfluidic channels to a significantly lesser degree than SCD RBCs (n=3 paired samples). Results are shared in a new **Supplementary figure 6A**:**

This new Supplementary figure and caption show RBCs from sickle cell disease blood samples are more significantly adhesive than healthy control counterparts:

"Supplementary figure 6. Repeatable brightfield microscopy analysis shows healthy control RBCs are less adhesive than SCD RBCs on laminin surfaces. (A) An estimation plot shows healthy control (AA) RBC samples adhere to laminin-coated microfluidic channels to a lesser degree than time-matched SCD (SS) RBC samples (n=3 paired samples, *p<0.05 by paired t test)."

This additional work is also referenced in the main manuscript:

*"Size and circularity metrics are generated as outputs, demonstrated here with heterogenous cell populations, including small/dense platelets (Figure 5A) and biconcave-shaped RBCs (Figure 5B, **Supplementary figure 6**)."*

3. In the new supplementary figure 11, below, **"ICLOTS analyses results serve as effective biomarkers congruent with findings from successful clinical trials"**, in response to the reviewer's request for more sampling we performed four additional experiments demonstrating a reduction in occlusion after crizanlizumab treatment, analyzed with the multiscale microfluidic **accumulation and occlusion** application. from **n=4 additional SCD samples, for a total of n=5 SCD samples**. Our data demonstrates that crizanlizumab treatment decreases the degree of CD45+ white blood cell and CD41+ platelet occlusion in 60% of treated samples.

“Supplementary figure 11. iCLOTS analyses results serve as effective biomarkers of pathophysiology congruent with findings from successful clinical trials. In experiments investigating the accumulation of CD41+ platelets and CD45+ white blood cells in a microvasculature-on-a-chip model (n=5 experiments), a ratio of microchannel occlusion without crizanlizumab treatment to microchannel occlusion with crizanlizumab treatment greater than 1 indicates that crizanlizumab is reducing the rate of microchannel occlusion within the microfluidic devices. Using the iCLOTS microfluidic device-scale accumulation and occlusion application, we find that in n=5 sickle cell disease patient samples, the ratio of (A) CD41+ platelet occlusion and (B) CD45+ white blood cell occlusion for treatment without crizanlizumab to treatment with crizanlizumab indicates reduction in occlusion (ratio > 1) in 60% of samples. (C) Ataga et al. (New England Journal of Medicine, 2016) found a similar lowered percentage rate of crises per year (1.63 with high-dose crizanlizumab treatment as compared with 2.98 with placebo, indicating a statistically significant 45.3% reduction) and uncomplicated crises per year (1.08 with high-dose crizanlizumab treatment as compared with 2.91 with placebo, indicating a statistically significant 62.9% reduction.)”

- **We wholeheartedly agree with the Reviewers that increased patient sampling would increase the impact of our manuscript, and therefore have added an additional nine single cell tracking experiments, three adhesion experiments, and four accumulation and occlusion experiments. We augment our existing statistically significant experimental findings within the manuscript with additional single cell tracking application experiments (n=10 iron deficiency anemia samples with matched controls), adhesion application experiments (n=3 sickle cell disease samples with matched controls), and microfluidic accumulation/occlusion experiments (n=5 sickle cell disease samples). These new experiments demonstrate that every iCLOTS image processing application can be used to process large amounts of data in order to produce statistically significant, clinically-relevant findings, per the Reviewers’ suggestions. These new statistically significant findings show (1) iron deficiency anemia RBCs are significantly smaller and less deformable than matched healthy controls, (2) SCD RBCs are significantly more adhesive than healthy matched controls, and (3) crizanlizumab reduces the CD45+ WBC and CD41+ platelet occlusion in SCD samples in a microvasculature-on-a-chip model. Using iCLOTS, users may efficiently analyze any number of samples collected.**

(3) A need to replicate previous findings from other literature and other labs

The Reviewers felt it would be useful if we took existing data from the literature, especially from other lab groups and other experimental workflows, to demonstrate that iCLOTS is effective in extracting useful information, including replicating the results of these studies. We thank the reviewers for valuable feedback on a pathway to demonstrating iCLOTS analysis veracity in a variety of microfluidic systems and cell types/multicellular structures not previously tested in iCLOTS.

- We appreciate the Reviewer request's to replicate previous findings as we feel it is an excellent opportunity to showcase iCLOTS' utility, versatility and potential for impact. In response, we have performed four additional experiments using Supplementary Movie files from previously published *Nature Communications* manuscripts. Using iCLOTS software to analyze these additional data files, we were able to recapitulate key findings from each of four manuscripts' data analyzed, including in other cell types and multicellular structures than those found in blood.
- We have validated experimental findings recapitulating the original conclusions of these four publications using videomicroscopy data provided in the papers themselves in a new Supplementary figure 13:

Supplementary figure 13. Data generated from iCLOTS recapitulates key findings from videomicroscopy of microfluidic experiments taken from previously published *Nature Communications* publications. (A) The iCLOTS single cell tracking application counts cells in motion as they move away from (control solution) or towards (chemokine solution) a chemokine. As published in *Boneschansker et al., 2014*, using Supplementary movie data downloaded directly from that paper via the *Nature Communications* website, we also find leukocytes in suspension migrate preferentially towards chemoattractant fMet-Leu-Phe (fMLP), that complement component 5a (C5a) induces both chemoattraction and repulsion in equal proportions in leukocytes, and that stromal cell-derived factor 1 (SDF-1) primarily repulses lymphocytes. (B) iCLOTS brightfield microscopy adhesion application counts coral polyps over time within Supplementary movie data monitoring a coral-on-a-chip microfluidic system, also indicating potential for use with other cells and/or multicellular structures. As published in *Shapiro et al., 2016*, we also find a gradual increase in salinity applied over time causes coral polyps to separate from each other. (C) iCLOTS accumulation and occlusion application quantifies fluorescence microscopy signal from Supplementary movie videomicroscopy data of a microfluidic device designed to recreate the circadian system. As published in *Gagliano et al., 2021*, we also find that by controlling the timing, period, and frequency of metabolic perturbations, circadian gene expression as detected by luminescence

oscillates over time. (D) iCLOTS accumulation and occlusion application quantifies fibrin deposition shown in Supplementary movie videomicroscopy of a specialized microfluidic device. As published in Jain et al., 2016, we also find that fibrin deposition from whole blood onto a microfluidic surface follows a sigmoidal trend.

- **Analysis of four videomicroscopy data sets from four separate microfluidic device designs monitoring multiple cell types validates that iCLOTS can be used to accurately analyze a wide range of experimental assays.** We reference the ability of iCLOTS software analysis to recreate other published findings in the main manuscript:

iCLOTS was designed around large data sets from multiple research groups and has been used to recreate key findings from previously published studies⁷⁵⁻⁷⁸ (Supplementary figure 13) in order to verify that the algorithms applied work consistently for a variety of experimental set ups, imaging parameters, and additional cell or multicellular structure types.

- These new experiments using published *Nature Communications* data indicate that using iCLOTS, users can indeed analyze their own unique microscopy images and videos, which may be taken from any-dimension microfluidic devices and multiple cell types and multicellular cell structures. **Per the Reviewers' suggestion, we have used iCLOTS to recreate key findings from other labs through investigation of four diverse microfluidic experiments** characterizing leukocyte migratory signatures, coral polyp separation under stress, circadian rhythms, and fibrin deposition dynamics.

(4) A need to provide more benchmarking against gold standard measurements

Reviewer 4 had requested that we obtain a matching dataset to verify the accuracy of the analysis algorithms, and suggested specifically that we compare cell size measurements from our microfluidic approaches to other measures of cell size or volume.

- We appreciate the Reviewer's suggestion, as it is an excellent opportunity to benchmark iCLOTS results. **We have performed the requested experiment comparing cell size measurements from the iCLOTS single cell tracking application with mean corpuscular volume (MCV) measurements taken from a complete blood count (CBC),** long understood to be the gold standard for RBC size. We find a **strong relationship between iCLOTS-generated cell area measurements and MCV,** presented in a new Supplementary figure 4:

“Supplementary figure 4. Metrics calculated by iCLOTS correspond with clinical gold-standard measurements. Mean RBC area (pixels) for nine patient blood samples meeting the criteria for iron deficiency anemia were compared to mean corpuscular volume (MCV) measurements taken by complete blood count (CBC). An increase in cell area measurements corresponds with an increase in MCV.”

- We have referenced this new experiment within the main text under the **Single cell tracking workflows** heading:

“iCLOTS-produced cell area values correspond with gold-standard clinical blood count (CBC) measurements of mean corpuscular volume (MCV), demonstrating result veracity (Supplementary figure 4).”

- **As requested by the Reviewers, comparison of iCLOTS-generated size metrics of RBCs are benchmarked against the gold standard for RBC size, MCV, therefore demonstrating that iCLOTS quantitative metrics do in fact correspond to clinical gold standard values.**

(5) Concern with implementation of constant flow rate and associated limitations

Reviewer 4 expressed concerns that results from pressure-driven vs. constant flow may affect result conclusions, which we can easily clarify and we apologize for the confusion.

- **The experimental methods presented in this manuscript are designed to give readers a sense of how the software can enable their own experiments but are not necessarily meant to be scientifically conclusive.**
- **However, for all experimental assay concerns presented, we demonstrate that iCLOTS has been designed to adapt to a wide range of experimental devices and configurations, and as such, software users can decide how best to use iCLOTS to meet their goals.** In an effort to guide the user, we have made numerous updates to our online and in-software help guide/documentation.
- For cell suspension velocity and cell adhesion experiments, a syringe pump was used to drive flow at a fixed rate to apply a constant physiologic shear stress throughout the duration of the experiment. Syringe pump driven flow was primarily used for the occlusion and deformability experiments for simplicity and ease of use. Small sample sizes also limited the use of larger pressure control systems. “Bypass” channels were included in the microfluidic device design for each of these systems to minimize changes in pressure drop in our ROI over the duration of the experiment. However, other microfluidic experiments we describe that were used in conjunction with iCLOTS indeed HAVE utilized flow driven by constant pressure as opposed to constant flow (see next point below)
- iCLOTS has been designed to adapt to a wide range of experimental workflows. **Select experiments (e.g. quantification of blood velocity in deoxygenated sickle cell disease whole blood samples, Figure 5A) were performed using pressure-driven flow, indicating iCLOTS analysis easily adapts to either syringe pump or constant pressure perfusion.**

Reviewer 4 also presented concerns with constant perfusion use specifically in our single cell tracking and accumulation and occlusion assays.

- **The previously-described microfluidic device used in our single cell tracking assays has large “bypass” channels on either side of the branching channels that prevent significant changes in pressure or potential clogging of the device,** which would affect velocity measurements. Additionally, experimental replicates are kept short (<2 minutes) to further prevent clogging or the buildup of adhesive factors that may occur over time.

We feel a significant contributor for this concern is that we did not sufficiently describe the assay used and we apologize for the confusion. In the text we state that *“In this assay, a greater cell velocity during transit of a microchannel smaller than the diameter of the cell indicates increased deformability.”* We appreciate the reviewer’s concerns and have therefore edited the text (please see 1A above) to emphasize the manuscript where the cell deformability microfluidic device is first described. **Accordingly, we have updated the supplementary methods to better describe the single cell tracking assay used and further address this Reviewer’s request to better integrate the main manuscript and supplement:**

“The cell deformability microfluidic device consists of a series of branching microfluidic channels surrounded by two large bypass channels designed to reduce potential for changes in pressure caused by microchannel clogging via slow-moving cells. Therefore, a range of cell velocities is indicative of a range of size and deformability phenotypes alone.”

- As shared above, syringe pump driven flow was primarily used for the occlusion and deformability experiments for simplicity and ease of use. **Small sample sizes associated with patient sample availability also limit the use of larger pressure control systems. However, the effect of heterogenous distribution of accumulation is a potential limitation, so at the reviewer’s suggestion we have indicated this in the Supplementary methods for the occlusion/accumulation assays:**

“Blood was perfused using constant flow via a syringe pump (PhD Ultra, Harvard Apparatus) into devices for 28 minutes. In branching microfluidic devices such as the device presented here, flow distribution may change over time due to the heterogenous distribution of accumulation.”

- Select experiments (e.g. quantification of blood velocity in deoxygenated sickle cell disease whole blood samples, Figure 5A) were performed using pressure-driven flow, indicating iCLOTS analysis is suitable for either syringe pump or constant pressure perfusion.
- **We appreciate the reviewer’s questions on considerations and limitations of constant perfusion vs. pressure-driven flow, so have updated the online and in-software user guide/documentation with appropriate guidance:**

“iCLOTS has been shown to produce accurate, reliable analyses of both constant perfusion (syringe pump) and pressure-driven flow across a range of microfluidic, flow-based experiments. While pressure-driven flow is more physiologically relevant, users may find they are limited by equipment availability or small sample sizes, or experimental set up may necessitate the greater simplicity or ease-of-use of constant perfusion systems. Users should carefully consider the importance of physiological relevance in their assays. If constant perfusion is used, consider designing microfluidic devices with large bypass channels to prevent significant changes in pressure from channel clogging.”

Concerns from Reviewer 1

In this manuscript Fay and colleagues describe the iCLOTS software toolbox they have developed for quantitative analysis of a multitude of static and dynamic experimental approaches that are used to describe hemostatic processes. While many of these analyses are already possible using common scientific imaging analysis programs such as ImageJ, such methods are often not standardized, depend on specialized plugins or writing of dedicated macros and require significant user intervention before meaningful data can be extracted. Specialized software packages such as iCLOTS that combine many common hemostatic analyses in a user-friendly environment are therefore a welcome contribution to our field. The purpose of this manuscript is to showcase the capabilities of iCLOTS by analyzing a number of datasets of adhesion of cells or proteins to adherent surfaces or microfluidic channels. Robustness of their analyses is demonstrated using sensitivity analysis and comparisons with manual quantification methods. The paper is well written and easy to follow, the

methods are described in sufficient detail. I only have a number of minor comments that mostly relate to usability and interface of the iCLOTS software.

- We thank the reviewer for their comments about the innovation and usefulness of iCLOTS.

To get a feeling for its functionality I have tried to use the iCLOTS program using some of the test data sets that were supplied by the authors. I found that the user interface does not scale well with the window size of the iCLOTS program. When images or videos are displayed, they remain fixed in the same small size even when stretching the window to cover the entire screen. At such limited image size, some of the object detections are difficult to discern, for instance the platelets adhered to collagen/fibrinogen matrix, and assessing whether changes in input parameters enhance or decrease object detection becomes difficult.

- As a core goal of iCLOTS is to be optimally useful to our user base, we thank the reviewer for trying the software and providing usability feedback. As shared in the manuscript:

“iCLOTS is shared with the greater research community with dedicated channels for feedback such that over time, as researchers contribute their own needs and workflows, the software will continue to develop and improve.”

We also provide specific contact information in the supplement within the section **Software access**.

- iCLOTS provides guidance within the software and online user guide that states: *“iCLOTS currently does not have a zoom function, but this is planned for a later release. In the meantime, if your data is relatively low-magnification, we suggest cropping a small region of interest using the video processing tools and testing parameters on that image, then applying the same parameters to the larger image.”*
- **However, we agree with Reviewer 1’s suggestion to scale the interface such that as window size increases image sizes increase as well would be a valuable improvement to our software, and therefore this and other changes (see below) have been implemented and packaged into a new version iCLOTS 0.1.1, available at <https://iCLOTS.org/software>. Updated source code is available at <https://www.github.com/iCLOTS>.**
- Similar to the process shown here, as we receive usability feedback, we will continue to update iCLOTS and publish new versions to our website and Github accounts.

On page 4 the authors state: “The original image and the image as analyzed displays in the center of the window, with changes in inputs of parameters updating in real time, allowing users to fit algorithms to their specific set of data (Figure 2D).” On my setup (PC with Windows 10) the iCLOTS program did not update detections setting in real time. For instance, for the detection of platelets adhered to collagen or fibrinogen, changes in the “Maximum diameter (pix)” or in the “Minimum intensity (a.u.)” parameters did not automatically update the object detection, but only when first clicking “Set detection parameters” and then select “Dark cells on light background”. Or in case I loaded multiple images at the same time, object detection was updated after flicking from one image to the other and back. This makes setting the right input parameters a tedious process. Same for object detection in videos, this only updated when going one frame forward (and then back), or when setting detection parameters and selecting dark/light cells on light/dark background.

- Our team has addressed event handling such that changes in parameters update immediately (without going from one image to another and back) after additional testing on Windows 10. We suspect some issues with changes in parameter values not appearing to register were due to (1) iCLOTS design as a robust product, i.e. small changes in parameters do not produce significant changes in numerical results,

and also (2) the small window size, which has been addressed in new iCLOTS version 0.1.1. Again, we thank Reviewer 1 for their usability feedback.

In video analyses (for example the transient adhesion time analysis) I found it very difficult to scroll through the frames. Only a very small sliding bar is present with which one can scroll through the frames but this is very insensitive and goes in steps of 6 frames. Please add buttons or some sort of explanation how to scroll through the video frame by frame.

- We thank the reviewer for this suggestion and accordingly, **we added keyboard controls such that for all applications displaying multiple images or video frames, when the application window is open and active users can scroll left or right one frame at a time using <Left> and <Right> keyboard arrows, respectively.** This feature is available in new iCLOTS version v0.1.1.
- In addition, we have added the following guidance on using the keyboard to manipulate the images displayed to the caption of Supplementary figure 1:

*“Here, a time course series of images is presented and can be accessed through a scroll bar beneath the analysis images. **Users may scroll through frames using <Left> and <Right> arrow keys.**”*

- We have also updated the online and in-software user guide/documentation.
- **By adding scalable windows, keyboard control of image series, and updated event handling, and by performing additional testing, the new version of iCLOTS packaged in response to reviewer feedback is an improved product.** As we receive additional feedback from users, we are committed to continuing to update iCLOTS, and as such usability and functionality will continue to increase over time.

Page 8, paragraph describing case 3: Figure 4F should be Figure 5F I presume?

- Yes, we thank the reviewer for noting this error. The manuscript has been corrected.

Neutrophil adhesion (Figure 5E) is quantified in numbers of neutrophils adhered and their transient adhesion times. Is it possible to analyze neutrophil crawling (motility and/or directionality, extravasation) using iCLOTS? This would leverage iCLOTS for applications that study transendothelial migration of various immune cells.

- We appreciate the Reviewer’s suggestion for an additional potential use case of iCLOTS applications. Yes, our single cell tracking application is already suitable for this task. The single cell tracking application provides velocity measurements and spatial coordinates for individual cells, including neutrophils. We have updated the online and in-software user guide/documentation to suggest this potential application:

“A potential use case for this application could be to analyze neutrophil crawling (including motility and/or directionality, and/or extravasation).”

- We appreciate all suggestions to improve iCLOTS. **As additional users provide feedback, the iCLOTS user guide and software functionality will continue to grow over time.**

Concerns from Reviewer 2

The authors present a software suite for analyzing time-dependent cell data, especially in microfluidics devices and blood related studies. Currently, most dynamic cell studies require custom code development and manual steps during analysis, and the authors' software, iCLOTS, aims to provide a simple to use solutions package. Overall, the concept is similar to popular open source software such as ImageJ and FIJI, but with emphasis on time dependent microscopy data that are common in blood related studies. There is strong merit to having such software and it can benefit many users.

- We thank the reviewer for their comments about the potential for iCLOTS to benefit many users.

Some comments to consider:

1) Integrating iCLOTS as part of ImageJ can leverage the many existing plugins available for ImageJ. This would seem to improve data analysis capacity for end users and is consistent with the central goals stated by the authors.

- ImageJ is an excellent general-purpose image analysis software with many successful associated projects. **However, we were inspired by other open-source image analysis software** with their own specialized functions, e.g. Ilastik for image segmentation or CellProfiler for advanced morphology metrics, **to create a single standalone product for a specific purpose (i.e. a focus on tracking and time course experiments)**. In this way, users do not need to perform multiple downloads and/or manage multiple plug-ins. In addition to our image processing applications, iCLOTS also incorporates a machine learning (ML) application designed to interpret numerical data (e.g. the numerical data produced by iCLOTS), a function separate from the stated goals of ImageJ. As we continue to grow and transform iCLOTS, **we feel that a standalone product or series of standalone products will be better suited for our targeted mission of facilitating analysis of clinical and/or diagnostic-focused assays with ML-augmented workflows.**

2) While this software can be very helpful especially to those with less analytics experience, this study in its current form appears to be primarily generating a collection of existing tools along with tailoring them toward specific applications in analyzing blood. The demonstration of specific applications in the case studies is intriguing and can significantly raise the impact of this study.

- We appreciate that the reviewer feels our software could be useful to our target audience, researchers with less analytics experience and/or no computational expertise. To our knowledge, no easy-to-use, adaptable, ML-augmented analytical techniques for methods incorporating fluid flow, dynamic adhesion, and/or commercially-available or novel microfluidics have been made widely available to the greater research community.

However, currently there is limited depth to the case studies and the patient sampling is low. The impact would increase if novel insights can be generated and validated, e.g. validation of specific biomarkers rather than suggestive findings, which will require larger sampling and patient numbers and addressing potential confounding factors. As presented, findings are only suggestive of potential applications, but without validation of new biomarkers or discovery of novel mechanistic insights. While this may require significant effort, it will raise the impact and novelty of the paper, especially if a new biomarker is validated and can be immediately used clinically or if a new mechanistic insight about a disease mechanism is discovered.

- We thank the reviewers for this point. That said, our primary purpose of this manuscript is to introduce a new software tool to multiple scientific and researcher user groups, as other similar *Nature Communications* publications have done, including the recent manuscripts describing B-SOiD, an

unsupervised algorithm for identification and fast prediction of animal behaviors (Hsu et al., *Nature Communications*, 2021) and BUNDLE, a tool for inferring localization of proteins (Crook et al., *Nature Communications*, 2022). We were also inspired by the manuscripts introducing the open-source software projects ilastik (Berg et al., *Nature Methods*, 2019) and CellProfiler (Carpenter et al., *Genome Biology*, 2006), which followed a similar approach of introducing a tool then providing a brief case study designed primarily to demonstrate utility. **Therefore, the major innovation of this paper is an interactive toolkit that addresses a need for a field stymied by a lack of analytical tools for innovative, physiologically-relevant assays of any design, with a special focus on methods incorporating fluid flow, dynamic adhesion, and/or commercially-available or novel microfluidics.** To that end, we've created a piece of standalone, free software that democratizes use of well-validated image analysis and machine learning algorithms for all end-user biomedical researchers who would benefit from advanced computational methods.

- However, we wholeheartedly agree with the Reviewers that validation of a key biomarker that provides novel mechanistic insight would add to the impact and novelty of our manuscript, and **in response to this suggestion, we have validated a key biomarker for SCD pathophysiology using our iCLOTS software** by collecting additional data that shows a reduction in CD41+ platelet and CD45+ WBC occlusion of *in vitro* microchannels after treatment with the newly FDA-approved drug crizanlizumab. Our data validates that platelet and WBC occlusion in microfluidic channels can be used as an objective biomarker, enabled by iCLOTS, that correlates with clinical outcomes, specifically the efficacy of new sickle cell therapies like crizanlizumab. Importantly, our new data is congruent with clinical trial results published in the *New England Journal of Medicine*, which led to the FDA approval of crizanlizumab for sickle cell disease. That paper demonstrates clinical efficacy of the drug while our data demonstrates that microfluidic occlusion using iCLOTS can serve as an objective biomarker for the clinical endpoints in that paper.
- Specifically, for this experiment, we have performed iCLOTS analysis using the multiscale microfluidic accumulation/occlusion application with four additional sickle cell disease patient samples (total n=5) with and without crizanlizumab treatment, **presented in a new Supplementary figure 11:**

“Supplementary figure 11. iCLOTS analyses results serve as effective biomarkers of pathophysiology congruent with findings from successful clinical trials. In experiments investigating the accumulation of CD41+ platelets and CD45+ white blood cells in a microvasculature-on-a-chip model (n=5 experiments), a ratio of microchannel occlusion without crizanlizumab treatment to microchannel occlusion with crizanlizumab treatment greater than 1 indicates that crizanlizumab is reducing the rate of microchannel occlusion within the microfluidic devices. Using the iCLOTS microfluidic device-scale accumulation and occlusion application, we find that in n=5 sickle cell disease patient samples, the ratio of (A) CD41+ platelet occlusion and (B) CD45+ white blood cell occlusion for treatment without crizanlizumab to treatment with crizanlizumab indicates reduction in occlusion (ratio > 1) in 60% of samples. (C) Ataga et al. (New England Journal of Medicine, 2016) found a similar lowered percentage rate of crises per year (1.63 with high-dose crizanlizumab treatment as compared with. 2.98 with placebo, indicating a statistically significant 45.3% reduction) and uncomplicated crises per year (1.08 with high-dose crizanlizumab treatment as compared with 2.91 with placebo, indicating a statistically significant 62.9% reduction.)”

- We have also referenced the new Supplementary figure 11 within the main manuscript text:

*“The iCLOTS accumulation application automatically generates a map of complex microfluidic device dimensions (Figure 6B). We expectedly observed that both CD41+ platelets and CD45+ WBCs within whole SCD blood treated with crizanlizumab occlude microfluidic channels less than untreated SCD blood samples (Figure 6C) **at a lowered percentage rate similar to completed clinical trials (Supplementary figure 11)**”⁶⁹.*

Where ⁶⁹ indicates the clinical trial referenced:

Ataga, K. I. et al. Crizanlizumab for the Prevention of Pain Crises in Sickle Cell Disease. *New England Journal of Medicine* 376, 429-439, doi:10.1056/NEJMoa1611770 (2016).

- These new experiments indicate that using iCLOTS, **per the Reviewers’ suggestion, we can indeed validate new biomarkers, and in this case we show a reduction in CD41+ platelet and CD45+ WBC occlusion of microchannels after treatment with crizanlizumab** congruent with clinical trial findings published in the *New England Journal of Medicine*.

3) The authors emphasize the need for simple to use quantitative tools for characterizing cell features in time-dependent and microfluidic experiments and how their software addresses this need. It would be useful if the authors take existing data from the literature, especially from other lab groups and other experimental designs, to demonstrate effectiveness of their software in extracting useful information, including replicating some results of those studies. This will help demonstrate ease of use, amount of manual customization required or to be expected from end users, and applicability for other device designs and biomedical applications beyond the case studies here.

- **We appreciate the Reviewer’s request to replicate previous findings as we feel it is an excellent opportunity to showcase iCLOTS’ utility, versatility and potential for impact. In response, we have performed four additional experiments using Supplemental Movie files from previously published *Nature Communications* manuscripts. Using iCLOTS software to analyze these additional data files, we were able to recapitulate key findings from each of four manuscripts’ data analyzed, including in other cell types and multicellular structures than those found in blood.**
- We have validated experimental findings recapitulating the original conclusions of these four publications using videomicroscopy data provided in the papers themselves in a new Supplementary figure 13. We

include the relevant caption text describing recreation of four key results from four manuscripts with four separate experimental workflows below:

Supplementary figure 13. Data generated from iCLOTS recapitulates key findings from videomicroscopy of microfluidic experiments taken from previously published Nature Communications publications. (A) The iCLOTS single cell tracking application counts cells in motion as they move away from (control solution) or towards (chemokine solution) a chemokine. As published in *Boneschansker et al., 2014*, using Supplementary movie data downloaded freely from the Nature Communications website, we also find leukocytes in suspension migrate preferentially towards chemoattractant fMet-Leu-Phe (fMLP), that complement component 5a (C5a) induces both chemoattraction and repulsion in equal proportions in leukocytes, and that stromal cell-derived factor 1 (SDF-1) primarily repulses lymphocytes. (B) iCLOTS brightfield microscopy adhesion application counts coral polyps over time within Supplementary movie data monitoring a coral-on-a-chip microfluidic system, also indicating potential for use with other cells and/or multicellular structures. As published in *Shapiro et al., 2016*, we also find a gradual increase in salinity applied over time causes coral polyps to separate from each other. (C) iCLOTS accumulation and occlusion application quantifies fluorescence microscopy signal from Supplementary movie videomicroscopy data of a microfluidic device designed to recreate the circadian system. As published in *Gagliano et al., 2021*, we also find that by controlling the timing, period, and frequency of metabolic perturbations, circadian gene expression as detected by luminescence oscillates over time. (D) iCLOTS accumulation and occlusion application quantifies fibrin deposition shown in Supplementary movie videomicroscopy of a specialized microfluidic device. As published in *Jain et al., 2016*, we also find that fibrin deposition from whole blood onto a microfluidic surface follows a sigmoidal trend.

- **Analysis of four videomicroscopy data sets from four microfluidic device experiments monitoring multiple cell types validates that iCLOTS can be used to accurately analyze a wide range of experimental assays.** We reference the ability of iCLOTS software analysis to recreate other published findings in the main manuscript:

iCLOTS was designed around large data sets from multiple research groups and has been used to recreate key findings from previously published studies⁷⁵⁻⁷⁸ (Supplementary figure 13) in order to verify that the algorithms applied work consistently for a variety of experimental set ups, imaging parameters, and additional cell or multicellular structure types.

- These new experiments using published *Nature Communications* data indicate that using iCLOTS, users can indeed analyze their own unique microscopy images and videos, which may be taken from any-dimension microfluidic devices and multiple cell types and multicellular cell structures. **Per the Reviewer’s suggestion, we have used iCLOTS to recreate key findings from other labs through investigation of four diverse microfluidic experiments** characterizing leukocyte migratory signatures, coral polyp separation under stress, circadian rhythms, and fibrin deposition dynamics.

The authors can also discuss if there are certain experimental designs that are required in order for the software to be immediately usable and the potential limitations.

- In the manuscript discussion, we share that iCLOTS can adapt to any-dimension microfluidic device or static system:

“iCLOTS algorithms and applications have all been designed to adapt to any microfluidic device or static system, independent of channel number, size, or dimensions, and is available as a standalone, easy-to-use product to any and all biomedical researchers.”

- In our new Supplementary figure 13 (above), we recreate several previously published *Nature Communications* findings, each which used a different microfluidic device not previously analyzed by iCLOTS, including devices with multiple channels in multiple orientations.

- Additionally, we share key limitations of iCLOTS in the manuscript discussion:

“However, analysis success depends on the quality of imaging data presented (Supplementary figure 12). Some level of noise or spurious features is to be expected with all high-throughput data analysis. iCLOTS is designed to label all analyzed imaging data with single-cell indices corresponding to numerical data, allowing users to assess if outliers are valid data points. If data is unacceptably noisy, it may still be less labor-intensive to correct computationally produced data than to perform manual analysis, which may be error-prone itself.”

- **During the course of iCLOTS development, both the experimental and computational teams have picked up a series of “tips and tricks” that would contribute to the immediate usability of the software, including information on potential limitations that address the Reviewer’s concerns.** These are shared in the online and in-software user guide/documentation. Please see a sample excerpt for the iCLOTS brightfield microscopy adhesion application from <https://www.iCLOTS.org/documentation>:

“Some tips from the iCLOTS team:

- *Computational and experimental methods:*
 - *The tracking methods used search for particles represented by image regions with Gaussian-like distributions of pixel brightness.*
 - *Analysis methods cannot distinguish between overlapping cells.*
 - *If cells are significantly overlapping, repeat experiment with a lower cell concentration.*
 - *Owing to the heterogenous appearance of certain cell types (e.g. the classic biconcave red blood cell shape, or the textured appearance of activated white blood cells), brightfield analysis may be challenging.*
 - *Consider using a fluorescent membrane stain coupled with our fluorescence microscopy adhesion applications if this does not conflict with your experimental goals, especially for WBCs/neutrophils.*

- *Choosing parameters:*
 - *Be sure to use μm -to-pixel ratio, not pixel-to- μm ratio.*
 - *Err on the high side of maximum diameter and low side of minimum intensity parameters unless data is particularly noisy or there's a large amount of debris.*
- *If you're unsure if what parameter values to select, run the analysis with an artificially high maximum diameter and low minimum intensity and compare indexed cells to the resultant metrics - for example, perhaps you see a cell typically has a diameter of "x" so you set maximum diameter slightly higher to exclude debris, and a cell typically has a pixel intensity of "y" so you set minimum intensity just below this to exclude noise.*
 - *The maximum diameter parameter can behave non-intuitively if set too high for the sample presented. If you cannot detect clear cells, try lowering this parameter.*
- *Output files:*
 - *Analysis files are named after the folder containing all images (.xlsx) or image names (.png)*
 - *Avoid spaces, punctuation, etc. within file names.*
 - *In use cases where several files are analyzed, individual sheets are named after individual files. These file names may be cropped to about 15 characters to prevent corrupting the output file. Please make sure individual files within a folder are named sufficiently differently.*
 - *Excel and pairplot data includes a sheet/graph with all images combined*
 - *Only use this when analyzing replicates of the same sample."*

Similar usability guidance exists for every application and will be updated over time as the iCLOTS experimental and development teams and new users provide feedback.

4) The filopodia tool is counting something similar to corners, but those may not be actual filopodia which are driven by certain molecular factors such as myosin X and fascin. A test that demonstrates the percentage of false positives/false negatives can be helpful.

- We have found what constitutes a filopodia to be subjective based on an researcher's objectives, and, potentially, biased. **A key benefit of iCLOTS is that criteria are applied objectively across an entire image or series of images, improving result repeatability and reliability.**
- **However, we appreciate the reviewer's suggestion that an experiment demonstrating the percentage of false positives and false negatives could be informative for users, and therefore we have added the requested calculations to an updated Supplementary figure 7.** We have split the original Supplementary figure 5 into an additional figure with standard sensitivity and precision calculations for a sample image (see below).
- We did not calculate specificity because any number of pixels not indicated as a filopodia could be considered a true negative, a key component of the standard specificity equation.

“Supplementary figure 7. Specialized iCLOTS filopodia counting application generates a single-cell resolution filopodia count for individual cells. (A) Number of filopodia per cell is quantified in an image of fluorescently stained platelets in a fraction of the time required for manual analysis. Data taken at 100x magnification, scale bars represent 50 μ m. (B) In this instance, iCLOTS analysis tends to overestimate number of filopodia as compared to manual analysis. (C) Sensitivity analysis shows analysis of the sample image performed with similar corner sharpness, or how distinct a filopodia must be from the main body of the cell, parameter values does not produce statistically significant changes in results ($p=0.22$ by ANOVA). Higher values of sharpness, indicating more permissiveness, result in higher numbers of filopodia detected. (D) Because manual filopodia characterization can be subjective, iCLOTS applies uniform criteria across an image, increasing reliability and repeatability of results. Here, as compared to a manual analysis performed by a trained hematologist, iCLOTS had a sensitivity of 95% and a precision of 90%.”

- This new experiment is referenced in the main text:

“Using this application, researchers can objectively apply criteria for filopodia detection (Supplementary figure 7).”

- Per the Reviewer’s suggestions for additional calculations, this new experiment performed in response verifies that using iCLOTS, filopodia counting tool results are indeed similar to manual analysis by a trained hematologist.

Concerns from Reviewer 3

On a general level, the main and unique result of this work is an overall software platform that integrates imaging, data analysis, feature extraction, machine-learning and interpretation tools through a guided step-by-step strategy. Inputs can be obtained from different microscopy or microfluidic sources, making the whole system very flexible and potentially useful for many users. In general, the whole work is excellent and has high potential impact.

- We thank the reviewer for their comments supporting the utility and potential for impact of iCLOTS.

As for the users, they are expected to be biomedical researchers. In the paper, the authors frequently mention the role of the user in configuring different parameters throughout the workflows and tend to state that the system is easy to use. This implies knowledge in several disciplines. I would recommend the authors to discuss whether such a system could also be accessible to clinical laboratories (also mentioned in the paper), what could broaden the scope of practical use of their system.

- As the reviewer has noted, we have mentioned potential for clinical laboratory use several times within the manuscript, including in our discussion:

*“Other specialized open-source tools¹⁰⁻¹³ and industry-based proprietary software typically require additional software, onerous complex calculations and/or coding/scripting on the user’s part, or are specific to a specific microfluidic device, which excludes many researchers and **clinical laboratories** from accessing and using those tools.”*

- **In response to the Reviewer’s suggestion to further consider clinical use, we have added additional discussion of iCLOTS use in clinical laboratories:**

“Manual analysis is also prone to bias that may contribute to reproducibility issues, especially if data is analyzed in an unblinded fashion, in a way that iCLOTS results are not. This potential to reduce bias and apply algorithms without need for computational expertise makes iCLOTS especially well-suited for use in clinical laboratories, where making key medical treatment decisions relies on robust, reproducible results. In this way, iCLOTS enables a semblance of standardization of data analysis for cellular microscopy data which is needed if these assays are to be used as clinical diagnostics and the regulatory processes thereof.”

- As a key goal of iCLOTS is to reach the widest possible range of users who would benefit from unbiased, completely automated analysis of cellular microscopy assays, we appreciate the Reviewer’s feedback that we should better explore what could broaden the scope of practical use of our system.

The work is truly original and robust and can have a scientific and practical impact. Compared to previous systems, iCLOTS is automated versus manual techniques, and is independent of imaging sources. It is also free and open access, compared to commercial systems. The work has presented four applications involving blood cells where analytical needs are highly relevant. All cases have been satisfactorily addressed and are describe in full detail. The conclusions are supported by these results and the discussions delivered. There are different techniques involved in this work and applied to four relevant cases. Not apparent flaws have been detected in the data analysis and interpretations appear correct. The different methodologies implied in this work seems appropriate within the state of the art. The main innovation lies in the overall strategy and how the different tools are nicely combined to build the system.

- We thank the reviewer for their comments supporting the robustness, innovation, and value in open access of iCLOTS.

I have two remarks that authors should discuss more precisely. They refer to the image quality. Authors mention a robust design around data sets from multi groups. The concept of robustness should be more clearly defined and the measures taken to make a robust design should be explained. In the same context, the dependence on the image quality is associated to “label all analyzed imaging data with single-cell indices corresponding to numerical data, allowing users to assess potential outliers”. This sentence is found unclear and unrelated to image quality.

- **In response to the Reviewer’s suggestion, we have clarified the concept of “robust” results** to define “robust” as proof that the algorithms we’ve adapted for use with cells are applied consistently across a variety of data sets and to clarify the utility of the single-cell indices:

“iCLOTS was designed around data sets from multiple research groups in order to verify that the algorithms applied work consistently for a variety of experimental set ups and imaging parameters. However, analysis success depends on the quality of imaging data presented (Supplementary figure 14). Some level of noise or spurious features is to be expected with all high-throughput data analysis. iCLOTS is designed to label all analyzed imaging data with single-cell indices corresponding to numerical data, allowing users to assess if outliers are valid data points.”

- Any new experimental or computational tool can require significant troubleshooting to implement. **Per the reviewer’s suggestions to add additional user guidance and in reference to their concern about image quality, we have created an additional Supplementary figure 14 that provides real-world examples of ideal and poor image quality, including suggested corrections to common image analysis issues, in order to help iCLOTS users quickly adapt to the analysis techniques now available to them:**

“Supplementary figure 14. Common data quality issues can be solved using iCLOTS experimental guidance or the iCLOTS suite of video editing tools in order to improve final feature quantification. The most common data quality issues the iCLOTS development team encounters are: (A) too high a cell density, which may be solved by reducing overall cell concentration; (B) a low signal-to-noise ratio, which may be solved by adjusting imaging data contrast values; (C) low signal from immunofluorescence staining, which may be solved by calibrating staining procedures and microscopy settings; (D) too low a resolution for quantifying morphological features, which may be solved by choosing a higher-magnification microscope objective; or, (E) too high a resolution for high-throughput data analysis, which may be solved by reducing file resolution and/or length.”

The core paper and the supplemental material are well written and provide enough detail on the methods. Qualified researchers could reasonably reproduce the results. The authors also offer public documentation on a web site.

- We thank the reviewer for comments on the quality of our work, our software, and our documentation (the online and in-software user guide/documentation referenced in this revision document).

Concerns from Reviewer 4

The authors have developed a suite of open source artificial intelligence-enabled software for analyses of blood cells in microfluidic and microscopy-based assays. They demonstrate the capability of their software by passing clinically relevant blood samples (whole blood or washed cells) through microfluidic devices mimicking the microvasculature. Analysis is performed on both capillary and venular sized rectangular channels.

The focus of the manuscript is to provide other researchers with access to advanced analysis tools for microfluidic studies of blood and other cell types.

Although I strongly support the goals of the authors I have a number of concerns. For wide use by researchers who are not experts in this type of analysis there needs to be more guidance on the practical use of the software and better integration of the main text with the Supplement.

- We thank the Reviewer for raising concerns that allow us to preemptively update our materials to address the needs of iCLOTS users. **We agree with this and other similar concerns presented by the Reviewer that users may need additional guidance when using iCLOTS, and in response have added numerous updates to the manuscript, software, and online and in-software user guide/documentation.**
- All iCLOTS applications are based on a similar layout, with parameters to adjust on the left, real-time analysis in the center, and options to run analysis and export results on the right. This intuitive format is detailed in main manuscript results section “*Standalone software is designed to balance ease-of-use with maximum functionality*”, in Figure 2, and Supplementary figure 1. Users can also watch a real-time demonstration of iCLOTS usage in Supplementary Video 1.
- **To assist the users in downloading and setting up iCLOTS, we have included a new **Software Use** section within the supplement providing additional instruction:**

iCLOTS has been designed as a standalone software to reach the widest range of users possible and for potential use in clinical environments. As such, no supporting software or software dependencies are required. No additional resources are needed to run the program. iCLOTS is installed simply by downloading the appropriate files. On Mac OS, users click the tar.gz distribution file to open, then click the .app file to start the software. On windows, users can click on the .exe file directly. iCLOTS initial version v0.1.0 is approximately 150 MB large, so may take 1-10 minutes to download, depending on internet speed. iCLOTS will take an additional 1-5 minutes to open, particularly for first time use. The development team has taken the necessary steps to identify ourselves as legitimate developers to Mac and Windows OS. Upon opening for the first time, you may receive messages alerting you that the software has been downloaded from the internet and/or that iCLOTS is a new piece of software. The source of the software will be attributed to Meredith Fay, lead developer. During testing, all software users received and accepted these messages with no negative effect to their computers.

- This information on software use has also been added to the online and in-software user guide/documentation.

Practical advice is missing for why specific choices were made for the microfluidic experiments. Why would one chose a fixed flowrate with a syringe pump vs constant pressure perfusion?

- **We appreciate the Reviewer’s concerns resulting from pressure-driven vs. constant flow and how this may affect result conclusions, which we can easily clarify and we apologize for the confusion.**
- The experimental methods presented in this manuscript are designed to give readers a sense of how the software can enable their own experiments but are not necessarily meant to be scientifically conclusive. However, **for each experimental assay concern presented by the Reviewer, we demonstrate that iCLOTS has been designed to adapt to a wide range of experimental paradigms and configurations, and as such, software users can feel confident that iCLOTS will fulfill their analysis needs for any number of experimental designs.**
- For cell suspension velocity and cell adhesion experiments, a syringe pump was used to drive flow at a fixed rate to apply a constant physiologic shear stress throughout the duration of the experiment. Syringe pump driven flow was primarily used for the occlusion and deformability experiments for simplicity and ease of use. Small sample sizes also limited the use of larger pressure control systems. Bypass channels were included in the device design for each of these systems to minimize changes in pressure drop in our ROI over the duration of the experiment.
- iCLOTS has been designed to adapt to a wide range of experimental workflows. **Select experiments (e.g. quantification of blood velocity in deoxygenated sickle cell disease whole blood samples, Figure 5A) were performed using pressure-driven flow, indicating iCLOTS analysis easily adapts to either syringe pump or constant pressure perfusion.**
- **We appreciate the reviewer’s questions on considerations and limitations of constant perfusion vs. pressure-driven flow. In an effort to guide the user in selecting the most appropriate flow paradigm for their assay, we have updated our online and in-software help guide/documentation with the following information:**

“iCLOTS has been shown to produce accurate, reliable analyses of both constant perfusion (syringe pump) and pressure-driven flow across a range of microfluidic, flow-based experiments. While pressure-driven flow is more physiologically relevant, users may find they are limited by equipment availability or small sample sizes, or experimental set up may necessitate the greater simplicity or ease-of-use of constant perfusion systems. Users should carefully consider the importance of physiological relevance in their assays. If constant perfusion is used, consider designing microfluidic devices with large bypass channels to prevent significant changes in pressure from channel clogging.”

What range of hematocrit levels to use in the feed reservoir for RBC suspensions and why?

Though iCLOTS works best when cells are distinct and do not overlap, **any cell hematocrit can be analyzed using iCLOTS, and as such the cell concentration chosen for any application is purely at the user’s discretion.** Our experiments were merely meant to be example case studies to illustrate iCLOTS’ capabilities. We empirically chose hematocrits to ensure that we could operate within a quantifiable dynamic range of the microfluidic devices for both healthy or untreated controls and experimental samples. For example, for the red cell adhesion experiments, we chose hematocrits between 0.1%-1% such that the final level of red cell adhesion for experimental samples did not fully saturate the channel or result in significant red cell overlap.

- **However, we appreciate the reviewer’s question as it brings to light key practical choices the user must make, and as such we feel users would benefit from guidance on choosing an appropriate**

cell concentration. We have updated the online and in-software user guide/documentation with this information:

"For all experiments involving quantification of single cell events, in our experimental and software testing we chose cell concentrations or hematocrits to ensure that we could operate within a quantifiable dynamic range of the microfluidic devices for both healthy or untreated controls and experimental samples. iCLOTS in its current iteration cannot distinguish between overlapping cell events. Typically we perform an initial experiment with a range of cell concentrations such that the most adhesive samples can adhere without overlap, then use this concentration for all future experiments."

- New Supplementary figure 14A also provides a visual example of choosing cell concentrations most appropriate for iCLOTS analysis:

“Supplementary figure 14. Common data quality issues can be solved using iCLOTS experimental guidance or the iCLOTS suite of video editing tools to order to improve final feature quantification. The most common data quality issues the iCLOTS development team encounters are: (A) too high a cell density, which may be solved by reducing overall cell concentration; (B) a low signal-to-noise ratio, which may be solved by adjusting imaging data contrast values; (C) low signal from immunofluorescence staining, which may be solved by calibrating staining procedures and microscopy settings; (D) too low a resolution for quantifying morphological features, which may be solved by choosing a higher-magnification microscope objective; or, (E) too high a resolution for high-throughput data analysis, which may be solved by reducing file resolution and/or length.”

When do you use bright field illumination vs fluorescence and why?

- iCLOTS has been designed to adapt to a wide range of experimental workflows. **The choice of brightfield illumination vs. fluorescence microscopy depends on the user’s experimental goals.** Brightfield microscopy does not rely on any type of cell labeling. We’ve found some stains can affect cell membrane properties, i.e. R18 appears to damage the RBC membrane. In experiments where simple count or simple movement is quantified, brightfield microscopy is typically sufficient.
- However, blood cells naturally have a heterogenous membrane appearance, which can affect area or other morphology measurements. To obtain the highest signal-to-noise ratio (e.g. the most apparent difference between imaging background and cell signal), we have stained cells or cell solutions with a stain indicating the cell membrane and taken images using fluorescence microscopy. The fluorescence microscopy adhesion assay also quantifies a secondary stain indicating some biological activity, providing users with an additional quantitative metric describing functionality of individual cells.
- **We appreciate the reviewer’s question on choosing microscopy modalities and feel users would benefit from guidance on choosing a unlabeled (brightfield) vs. labeled (fluorescence microscopy) approach, so have updated the online and in-software user guide/documentation with appropriate guidance:**

“Brightfield microscopy does not rely on any type of cell labeling. We’ve found some stains can affect cell membrane properties, i.e. R18 appears to damage the RBC membrane. In experiments where simple count or simple movement is quantified, brightfield microscopy is typically sufficient.”

Blood cells naturally have a heterogenous membrane appearance, which can affect area or other morphology measurements. To obtain the highest signal-to-noise ratio (e.g. the most apparent difference between image background and cell signal) we recommend staining cells or cell solutions with a stain indicating the cell membrane and using fluorescence microscopy. The fluorescence microscopy adhesion assay quantifies a secondary stain indicating some biological activity. Future version of iCLOTS will incorporate secondary “functional” quantification in additional applications.”

Why were all of the experiments run at room temperature when temperature has a significant effect on RBC deformability and cell physiology?

- We agree that temperature does have a significant effect on RBC deformability and cell physiology. After initial experiments comparing RBC quantitative results at room temperature and a physiological temperature of 37°C yielded no significant differences, we chose to perform our experiments at room temperature to simplify experimental setup.
- Several other similar manuscripts investigating RBC physiology in microfluidic devices (Man et al., *Lab on a Chip*, 2021; Qiang et al., *Lab on a Chip*, 2022), have performed RBC experiments at room

temperature, so in this way, we also wanted to illustrate a “typical” experimental set up. To help account for this and other variables that were not included in these models, all samples were performed in the same way, at room temperature, and were analyzed relative to a healthy donor or untreated control.

- **iCLOTS has been designed to adapt to a wide variety of experimental designs and conditions. If a user has a microscopy setup that controls temperature or oxygen levels, iCLOTS can accommodate these workflows as well.**
- The primary purpose of this manuscript is to introduce a new tool to a series of users by introducing each application and providing an experimental case study that demonstrates inputs, parameters, outputs, and potential use cases. Temperature as well as the inclusion of hypoxia would be excellent additional parameters to include in future studies.

For many of the examples given, there are other methods for obtaining a matching dataset to verify the accuracy of the analysis algorithms and interpretation of the microfluidic results. It would strengthen the paper to include comparison to “gold standard” measurements. For example: an advantage of the microfluidic approach is direct comparison between cell velocity and cell size but how does the cell size measurements compare with other measures of cell size or volume?

- We appreciate the Reviewer’s suggestion, as it is an excellent opportunity to benchmark iCLOTS results. **We have performed the requested experiment comparing cell size measurements from the iCLOTS single cell tracking application with mean corpuscular volume (MCV) measurements taken from a complete blood count (CBC), long understood to be the gold standard for RBC size. We find a strong relationship between iCLOTS-generated cell area measurements and MCV, presented in a new Supplementary figure 4:**

“Supplementary figure 4. Metrics calculated by iCLOTS correspond with clinical gold-standard measurements. Mean RBC area (pixels) for nine patient blood samples meeting the criteria for iron deficiency anemia were compared to mean corpuscular volume (MCV) measurements taken by complete blood count (CBC). An increase in cell area measurements corresponds with an increase in MCV.”

- We have referenced this new experiment within the main text under the **Single cell tracking workflows** heading:

“iCLOTS-produced cell area values correspond with gold-standard clinical blood count (CBC) measurements of mean corpuscular volume (MCV), demonstrating result veracity (Supplementary figure 4).”

- **As requested by the Reviewer, comparison of iCLOTS-generated size metrics of RBCs are benchmarked against the gold standard for RBC size, MCV, therefore demonstrating that iCLOTS quantitative metrics do in fact correspond to clinical gold standard values.**

The one case given of manual single cell RBC velocity measurements does not fully address the validity of the velocity algorithm (see below). The manuscript also lacks details on accuracy of the various algorithms and spatial resolution of the optical flow method.

- We appreciate the Reviewer’s concerns about the validity and accuracy of the various algorithms iCLOTS uses, as we do understand that providing users with incorrect or imprecise metrics could negatively affect the interpretation of their hard-won data, perhaps even leading to incorrect scientific conclusions. We apologize that our original manuscript and supplement lacked the key details necessary for users to feel confident in their analysis. **We take the validity of iCLOTS seriously, and as such have built iCLOTS’ analysis workflows upon well-validated, highly-cited computer vision and machine learning algorithms.** For each application, we share the original manuscript where the algorithm was described and the open-source library used to implement the algorithm. Please see the manuscript methods for our machine learning interpretation workflow, where citation 16 indicates the Python library used, citation 40 indicates the initial publication of the specific algorithm used, and citation 41 directs the user to a manuscript describing best use practices for this algorithm:
 - *“iCLOTS uses Python library scikit-learn¹⁶ to implement k-means clustering⁴⁰, a specific mathematical model understood to be a robust general-purpose approach to discovering natural groupings within high-dimensional data⁴¹.”*
- Similar text within the manuscript and/or supplement is also present for each of the image processing algorithms.
- **Where possible, iCLOTS analysis has been compared to manual results, with results found to be similar in each case:**
 - Supplementary figures 3A-B show that while some error is present within multiple manual analyses of the same data set, results from iCLOTS single cell tracking automated analyses of velocity are similar to expert hematologist analyses.
 - Supplementary figures 7B and 7D show that while counting filopodia can be subjective, results from iCLOTS filopodia counting application are not significantly different than expert hematologist analyses.
 - Supplementary figures 8B and 8E show that brightfield microscopy adhesion analysis and fluorescence microscopy adhesion analysis are not significantly different than expert hematologist analyses.
- **As with each of our image processing workflows, spatial resolution is single pixel**, with the goal of providing the most precise resolution possible.

Although there is a sensitivity analysis for some algorithm parameters the paper lacks data on reproducibility, i.e. analysis of repeated runs of the same sample. I would also expect a critical evaluation of the limitations of the different analysis techniques as well as the microfluidic approaches presented.

- **We appreciate the reviewer’s concern about reproducibility of iCLOTS analysis results and in response have performed additional experiments to confirm that each iCLOTS application is robust, e.g. repeated runs of the same sample produce similar results.**
- In the existing manuscript, using the **multiscale microfluidic cell accumulation application**, we find that **in n=3 experiments** using the same sickle cell disease patient whole blood sample, **all replicates show a similar pattern of accumulation** of CD41+ platelets and CD45+ white blood cells over time. From main manuscript Figure 6A:

“(A) Occlusion is calculated as percent of the region of interest that contains signal from individual color channels in fluorescence microscopy images. Accumulation of cell components on a surface is calculated as the change in fluorescence microscopy image signal area relative to the previous frame. Line graphs representing occlusion and accumulation over time are automatically generated. Here, CD41+ platelets and CD45+ white blood cells from sickle cell disease patient whole blood accumulate on an ibidi chamber device coated with collagen at a faster rate than healthy control whole blood (n=3 replicates).”

- **We have added additional repeatability experiments for each remaining category of image processing applications (single cell tracking, velocity profile, and adhesion):**
 1. **Additional single cell tracking experiments presented in Supplementary figure 3D show iCLOTS analysis of n=3 multiple experimental replicates of the same iron deficiency anemia RBC sample do not produce significantly different single-cell velocity results. Results are included in subfigure D in an updated Supplementary figure:**

The relevant caption text shows results of multiple experiments/videos of the same sample do not produce significantly different results:

“(D) iCLOTS results are ideally robust in the sense that they’re repeatable, i.e. similar experiments and analysis on the same sample should yield similar results. Single cell velocity measurements from three separate experimental trials with RBCs from the same iron deficiency anemia sample are not significantly different (n = 914, 829, and 931 RBCs, respectively, p=0.24 by ANOVA).”

We have referenced this additional experiment in the main manuscript:

“iCLOTS single cell tracking analysis methods are robust, **repeatable**, and reduce potential for manual error (Supplementary figure 3).”

- Additional **velocity profile and time course** experiments presented in Supplementary figures 5C-D show iCLOTS analysis of **n=4 sequential 2.5 second video segments** of sepsis whole blood transiting a channel, while similar overall, result in significantly different mean time course velocity and mean profile velocity values. These findings suggest users should take care in interpreting data from long microfluidic experiments, as artifact may be introduced by cells within a suspension sample settling, adhesive factor buildup, or other similar issues.

We share statistical analysis of our experimental findings in the relevant caption text:

*“Repeatability analysis performed on (C) profiles and (D) time course data for four sequential segments of a single 8 second video show potential for variability in results from microfluidic devices over time. Mean velocity profiles ($n=4$ segments, $*p<0.005$ via paired one-way ANOVA with Geisser-Greenhouse correction) and mean velocity time course series ($n=4$ segments, $**p<0.0001$ via ANOVA) were significantly different. Maximum velocity values per frame were not significantly different ($n=4$ segments, $p=0.48$).”*

In response to the reviewer’s suggestion, we find potential for artifact in regular microfluidic use may affect the experimental findings of our users. As such, we have added additional experimental guidance to the online and in-software user guide/documentation:

“Over the course of long microfluidic experiments, factors such as a buildup of adhesive factors on channel walls, cells within a suspension settling, or other variables may lead to artifacts within data. The iCLOTS team suggests plotting quantitative metrics with frame number as the x-variable to ensure results are reasonably consistent over time.”

3. Additional **adhesion** experiments shown in an updated Supplementary figure 6B-C show iCLOTS analysis of **$n=3$ paired experimental replicates** of the healthy control (B) and SCD (C) RBCs **do not produce significantly different cell density results**.

Statistical analysis is shared in the relevant caption text:

“Additional estimation plots show repeated runs of the same RBC sample for both (B) healthy controls (n=3 paired experiments) and (C) SCD samples (n=3 paired experiments) are not significantly different (p=0.60 and p=0.47, respectively).”

We have referenced this additional experiment in the main manuscript:

*“iCLOTS adhesion assays produce accurate, **repeatable**, and robust cell measurements in a fraction of the time required for manual analysis (Supplementary figures 6, 9).”*

- **Therefore, in response to the Reviewer’s concerns regarding repeatability of iCLOTS results, we have performed a series of similar experiments for each application to show that iCLOTS results are robust.** In addition to previous sensitivity analysis and comparison to manual results, we now show that iCLOTS results are repeatable, and, when necessary, have provided additional user guidance on considerations for microfluidic experimental use. **For each application, we complement previous sensitivity analysis and comparison to manual results, with Reviewer-suggested repeatability analysis in order to validate that iCLOTS produces reliable, robust results users can feel confident in using to draw scientific conclusions.**

Some indication of the minimal computer system requirements would be helpful.

- We thank the Reviewer for noting this crucial information about use was not previously included in the manuscript. Within the added **Software use** section within the supplement, the authors have added the following text:

iCLOTS requires a 64-bit operating system and a minimum of 8 GB RAM, with more suggested for large .avi file video sets.

- We have also updated the documentation available at <https://iCLOTS.org/documentation> with this information.

1. Figure 3 shows an example of single-cell tracking applications which provide high-resolution measurements of velocity relative to cell size and fluorescence intensity. The figure demonstrates “use of this application with a specialized microfluidic assay where velocity of a cell transiting a microchannel indicates relative cell stiffness.” a. I assume the microfluidic device referred to is the one shown in Figure 1 Panel D, and the device was perfused at a constant rate using a syringe pump as described in the Supplement (there needs to be better integration between the main text and the supplement)

- We apologize for the confusion: the device shown in Figure 1D is the branching microfluidic device used primarily in our multiscale microfluidic accumulation and occlusion assays. To make this distinction clearer to the reader, **we have changed the text describing the single cell tracking workflows to emphasize that this is a previously-described assay** and to more directly provide the associated citation. Please see our correction:

“We demonstrate use of the iCLOTS single cell tracking application with a previously described, microfluidics-based cell deformability assay designed to measure single cell mechanical properties³⁶. In this assay, a greater cell velocity during transit of a microchannel smaller than the diameter of the cell indicates increased deformability.”

Reference 36 refers to: Rosenbluth, M. J., Lam, W. A. & Fletcher, D. A. Analyzing cell mechanics in hematologic diseases with microfluidic biophysical flow cytometry. *Lab Chip* 8, 1062-1070, doi:10.1039/b802931h (2008).

- As the reviewer has assumed, the device is perfused at a constant rate using a syringe pump, as described in the supplement. To better integrate the manuscript and the supplement, we have included this detail in the main manuscript as well:

“Microfluidic devices were prepared as previously described and cells were perfused through the device at a constant rate using a syringe pump.”

We have also standardized the wording describing syringe pump perfusion within each series of single cell tracking experimental methods in the supplement. **These changes to the manuscript and supplement also address Reviewer 4’s concern that the manuscript and supplement should display better integration.**

b. Why is there such a wide distribution of RBC velocities as shown in Panel B and C with a constant perfusion rate, a symmetric branching pattern for the microfluidic channels and what appears to be a very low RBC density in the suspension? I would expect a narrow range of velocities with the distribution skewed towards low velocities (channels with high velocities should have a higher RBC flux and hence there should be a higher frequency at high velocities). Is the wide distribution of velocities for control RBCs due to an artifact introduced by the syringe pump or variability in dimensions of the microfluidic channels causing asymmetric distribution of flow? Or is it due to artifacts introduced by the analysis software?

- We thank the Reviewer for their concerns about the deformability device assays, as we feel a significant contributor for this concern is that we did not sufficiently describe the assay used, and we apologize for the confusion. In the text we state that *“In this assay, a greater cell velocity during transit of a microchannel smaller than the diameter of the cell indicates increased deformability,”* but did not properly reference the manuscript where the cell deformability microfluidic device is first described. We have therefore edited the text (please see 1A above) to emphasize the appropriate manuscript.
- **The previously-described microfluidic device used in our single cell tracking assays has large “bypass” channels on either side of the branching channels that prevent significant changes in pressure or potential clogging of the device,** which would affect velocity measurements. Additionally, experimental replicates are kept short (<2 minutes) to further prevent clogging or the buildup of adhesive factors that may occur over time.
- **Accordingly, we have updated the supplementary methods to better describe the single cell tracking assay used and further address this Reviewer’s request to better integrate the main manuscript and supplement:**

“The cell deformability microfluidic device consists of a series of branching microfluidic channels surrounded by two large bypass channels designed to reduce potential for changes in pressure caused by microchannel clogging via slow-moving cells. Therefore, a range of cell velocities is indicative of a range of size and deformability phenotypes alone.”

- As shared above, syringe pump driven flow was primarily used for the occlusion and deformability experiments for simplicity and ease of use. **Small sample sizes associated with patient sample availability also limit the use of larger pressure control systems. However, the effect of heterogenous distribution of accumulation is a potential limitation, so at the reviewer's suggestion we have indicated this in the Supplementary methods for the occlusion/accumulation assays:**

"Blood was perfused using constant flow via syringe pumps (PhD Ultra, Harvard Apparatus) into devices for 28 minutes. In branching microfluidic devices such as the device presented here, flow distribution may change over time due to the heterogenous distribution of accumulation."

- The quality of all results presented in the manuscript have been verified by comparison of iCLOTS-added cell indices within the processed imaging data to the corresponding numerical Excel data. Selected results from each application have also been compared directly to a manual count. Supplementary figure 3A shows a comparison of iCLOTS analysis to the analysis of three trained hematologists. While the figure is shared primarily to show the potential for error in manual analysis, e.g. all hematologists counting a different n of cells, we note that participant results were overall similar to iCLOTS analysis, and as such we feel confident there were no notable artifacts present during the computational analysis.
- **We appreciate the reviewer's questions on considerations and limitations of constant perfusion vs. pressure-driven flow, so have updated the online and in-software user guide/documentation with appropriate guidance:**

"iCLOTS has been shown to produce accurate, reliable analyses of both constant perfusion (syringe pump) and pressure-driven flow across a range of microfluidic, flow-based experiments. While pressure-driven flow is more physiologically relevant, users may find they are limited by equipment availability or small sample sizes, or experimental set up may necessitate the greater simplicity or ease-of-use of constant perfusion systems. Users should carefully consider the importance of physiological relevance in their assays. If constant perfusion is used, consider designing microfluidic devices with large bypass channels to prevent significant changes in pressure from channel clogging."

- With the Reviewer's guidance, we have clarified use of the deformability assay, which will undoubtedly benefit users who would also like to quantify mechanical properties of single cells in a high-throughput manner.

c. Why did you chose to show in the Table in Panel B an iron deficient RBC with half the pixel size and half the velocity of a control RBC? This reinforces the concept that iron deficient RBCs are both smaller and stiffer but in the graph in the same Panel there doesn't appear to be a relationship between iron deficient RBC size and RBC velocity. Since these iron deficient cells are stiffer I was expecting a negative relationship. I was expecting that RBC velocity would increase as RBC size decreased; hence as flow resistance decreased. At least that is what I would expect with a constant driving pressure microfluidic system. But the authors have used a syringe pump to perfuse the device, i.e. constant flowrate system. With a constant flowrate, RBCs regardless of their size and stiffness should be forced through the channels at the same velocity. A constant perfusion system is not sensitive to cell deformability.

- **We appreciate the Reviewer's concerns about the interplay of constant flow, size, and velocity in the deformability assay and apologize for the confusion caused by not properly describing the microfluidic device.** To clarify, we have added a description of the "bypass channels" that prevent pressure build up within the device such that RBC velocity does indicate combined size and deformability properties:

"The cell deformability microfluidic device consists of a series of branching microfluidic channels surrounded by two large bypass channels designed to reduce potential for changes in pressure caused

by microchannel clogging via slow-moving cells. Therefore, a range of cell velocities is indicative of a range of size and deformability phenotypes alone.”

- iCLOTS’ unique capability to provide multiple metrics (size, deformability) at single-cell resolution allows users to investigate how interplay of these multiple properties contribute to one cell phenotype.
- Sample data points shown in Figure 3B and other figures are chosen to demonstrate the metrics iCLOTS quantifies for each assay, with a secondary function of showing a reasonable range of quantitative values produced. Should readers be interested in seeing a more complete dataset, we have added the following text to the **Software access** section of the supplement:

“Sample data, including sample analysis, is available for all applications at <https://www.iCLOTS.org/software>. All data presented and detailed experimental protocols are available upon request.”

Additional text within this section provides contact information:

“Please contact us via the corresponding author, via the contact form on the iCLOTS website, or via the contact information available at <https://www.github.com/LamLabEmory> or <https://www.github.com/iCLOTS>.”

d. Supplement: Deformability assay microfluidic device preparation “Microchannel height was chosen such that cells must necessarily deform to transit the device. A microchannel height of 5 μm was prepared for RBCs and reticulocytes and a microchannel height of 13.8 μm was prepared for WBCs and cell lines.” But the width of microchannel was 5.9 μm which means the diagonal dimension for RBC and reticulocyte transit was 7.7 μm . Most normal size RBCs (~8.0 μm in diameter x 2.2 μm thick) would require minimal deformation to pass through these channels since they are not constrained to orient to the smallest dimension. Smaller iron deficient RBCs could pass without needing to deform at all. I am not convinced that the channels are not small enough to test RBC deformability based on their velocity. It also isn’t clear why the channels needed to be twice the size for WBCs since they also traverse the same capillaries as RBCs in vivo.

- The cell deformability microfluidic device height is limited by the constraints of microfluidic mask processing. As the reviewer noted, ideally the height of the mask would compress cells, particularly RBCs, to a greater degree. However, it’s difficult to reliably produce microfluidic masks smaller than 5 μm without significant variation in height across the surface of the mask. Small variations in height would affect cell velocity values, introducing artifact.
- **Rather than recreate a biological system, the cell deformability microfluidic device is designed to investigate biophysical properties, so does not necessarily need to recreate an *in vivo* capillary environment to create useful findings.** As such, the ideal height that identifies a range of stiffness values but does not lead to excess device clogging was determined experimentally for different cell types. iCLOTS single cell tracking algorithms have sub-pixel particle (i.e., cell) location/detection and tracking resolution and are capable of quantifying morphology and movement of several thousand cells within minutes. A key benefit of iCLOTS analysis is that it can detect small changes in deformability across a large cell population quickly and easily.

e. With a constant flowrate generated by the syringe pump, why did the iron deficient RBCs appear to have a lower mean velocity?

- Please see above discussion on the role of bypass channels in preventing pressure buildup throughout the device. **The reviewer may also be interested our previous publication exploring mechanical properties of iron deficient RBCs, which has been added to the manuscript as ref. 38.**

- Caruso, C. et al. Pathologic Mechanobiological Interactions between Red Blood Cells and Endothelial Cells Directly Induce Vasculopathy in Iron Deficiency Anemia. *iScience*, doi:<http://dx.doi.org/10.2139/ssrn.3981901>.

2. In Supplement Fig 3, iCLOTS single cell tracking is compared to manual analysis by three expert hematologists. No statistical analysis is provided and only the shape of the velocity distribution is given as a visual comparison. The discrepancy between manual measurements and iCLOTS, and between the three experts appears to be more than simple measurement error of a few micrometers due to locating individual RBCs. I suspect the differences reflect the challenge of identifying a discrete RBC from one frame to the next when there are multiple nearly identical RBCs in the channel. This is especially challenging at velocities shown in the distribution where the RBC are traveling from 2 to 6 times their length in a single frame. The data should be available to do a Bland-Altman analysis to detect systematic errors between experts and between experts and the tracking algorithms.<https://pubmed.ncbi.nlm.nih.gov/32456091/>

- We agree with the reviewer that understanding the source of discrepancy between manual and iCLOTS analysis is important. We have updated Supplementary figure 3A to include statistics from an ANOVA test demonstrating that results comparing iCLOTS and the three participants are significantly different ($p < 0.001$), but agree with the Reviewer that this may be influenced by simple measurement error.
- Therefore, **we have also included the suggested Bland-Altman plot analyzing agreement between iCLOTS and each participant**, presented as an updated Supplementary figure 3B. This plot shows that some differences between user analysis and iCLOTS analysis are indeed due to manual analysis fallibility. We include information on this updated statistical analysis within the Supplementary figure caption.

“Supplementary figure 3. iCLOTS single cell tracking application is robust, reduces analysis time, and produces data sufficiently detailed for machine learning analysis. (A) iCLOTS generates single cell velocity values for individual cells in a fraction of the time required for manual analysis. One two-minute video took three expert hematologists at least 50x as long to analyze. Results presented in this manuscript use at least 5 two-minute videos per condition presented. While iCLOTS performed similarly to manual analysis, we find inter-participant variability in manual analysis as well, indicating potential for error in analyzing large datasets by hand (difference * $p < 0.001$ by ANOVA). (C) Bland-Altman plot analyzes the agreement between iCLOTS and each participant. (B) Sensitivity analysis shows analysis of the sample video performed with similar minimum cell diameter parameter values produces similar results. Parameter values chosen intentionally to exclude all data points result in no detected cells. Parameter values chosen intentionally to include image noise results in a reduced number of cells due to quality restrictions imposed on data points. (D) iCLOTS single-cell tracking application also produces size values for individual cells. To assist users in parsing large, multi-dimensional data sets, a pair plot is generated. Automatically generated pairplots output with final data allow software users to quickly parse differences between analyzed conditions. (E) When all data points from all conditions are combined, k-means algorithms with a software-user input $n_{clusters}$ cluster these data points into groups describing numerically similar data points. Here RBCs are grouped into low- and high-velocity clusters based on optimized k-means calculations.”

- **This additional statistical analysis confirms the potential for error and bias in manual analysis of microfluidic data, per the reviewer’s suggestion.** We would also like to emphasize that manual analysis resulted in different n cell counts in all three hematologist analyses, a clear indication that manual analysis is susceptible to user error and bias.

Loss of RBC deformability in sepsis and activation of coagulation pathway are well established in animal models of sepsis and septic patients. The ability to study altered blood flow in microfluidic channels reflecting conditions found in venules could be an important clinically relevant tool. However as presented I have a number of concerns.

- We thank the Reviewer for their concerns about the sepsis assays and apologize for any confusion caused by insufficient experimental or computational guidance. In response to the concerns raised, **we have made a series of changes to the manuscript, supplement, and online and in-software guide/documentation to assist with use and interpretation our velocity profile application.** Please see the relevant changes made in direct response to each concern below.

a. In Figure 4 Panel A, testing sickle cells, the full width of the channel is shown (walls are visible although not labelled). In Panel B, the walls are not visible and although the channel images are same size the scale bars are different. It is not clear that the channels are 70 μm wide as described in the Supplement. Please clarify and redo images to show the walls. NOTE: velocity units in Panels A & B are incorrect. Please give explanation for the blue dots in the figure caption.

- We thank the reviewer for catching the incorrect units and scale bars. These errors have been corrected (please see updated Figure 5 below).
- We typically suggest that users choose a region of interest excluding channel walls. From our online and in-software user guide/documentation, we share that **excluding walls can reduce artifact:**

“In assays using microfluidics, small defects in microfluidic walls are often detected as cells or patterns of cells. The iCLOTS team suggests cropping all data taken using microfluidic systems to the channel area only.”

- Additionally, profile generation relies on automatically-calculated, linearly-spaced bins spanning the height of the channel. In applications where users are relying on creating an accurate velocity profile, selecting an region of interest outside of the channel will make wall velocities artificially slower or zero. **We agree with the reviewer that users would benefit from clarification of best practices when choosing a region of interest and have added the following guidance to the online and in-software user guide/documentation:**

“Calculating an accurate velocity profile relies on automatically-calculated linearly-spaced bins spanning the height of the channel. While cropping to a region of interest excluding channel walls is not required, for best profile results the team suggests doing so to avoid wall velocities that appear artificially slower.”

- However, cropping to a region of interest is not required, as you noted in panel A. **We have updated the figure as requested by the reviewer to include the channel walls for increased user clarity:**

- The cyan dots or lines represent tracked feature trajectories. We have updated the figure caption for increased user clarity:

(A) Cell suspension velocity applications rely on algorithms that find patterns within images, typically a cluster of cells, and algorithms that track these detected patterns from one frame to the next. To quantify cell suspension velocity, users must adjust window size (a region of interest in which a detected pattern is searched for in the subsequent frame), a minimum distance traveled, and an approximate feature/cell size for best quantification. Trajectories of individual cell patterns are labeled on the provided video data, seen here as cyan lines.

b. Channels perfused with “whole blood samples at shear rate of 350 sec⁻¹ chosen to recapitulate venous shear rate of a vessel of similar dimensions.” The reader is left with impression that this channel mimics a 70 μm diameter venule at a physiological relevant wall shear rate. But the channel is highly asymmetric with a depth (or height) of only 10 μm (Supplement: Cell suspension in sepsis velocity assay) which is much closer to sheet flow than to flow in a tubular vessel. Which wall was the shear rate calculated for? In vivo a 70 μm diameter venule has a much higher velocity than 500 μm/s so I assume the flowrate was set to match the wall shear rate along the 10 μm dimension (top or bottom of the channel, not the sides). Please clarify.

- The wall shear rate is calculated for the channel height rather than the width. The interpretation that this more approximates sheet flow rather than tubular flow is correct. We do suspect aggregation plays a role in the changed rheological properties of blood from sepsis patients (please also see below), so a height of 10μm effectively reduces analysis to a 2 dimensional rather than 3 dimensional process given the expected cross section of an RBC. Because iCLOTS is a manuscript written primarily to address computational tools, we will explore the role of aggregation in sepsis further in a later manuscript.
- However, **the underlying algorithms for the velocity profile application in the manuscript are robust and are also suitable for use with tubular-like, iso-symmetric flow. Please see our previous work published in the following manuscript:**

Szafraniec, H. M. et al. Feature tracking microfluidic analysis reveals differential roles of viscosity and friction in sickle cell blood. *Lab on a Chip* 22, 1565-1575, doi:10.1039/D1LC01133B (2022).

- To clarify this information to the reader, we have added the following text to the supplement, where the citation references the Szafraniec manuscript listed above:

“The presented microfluidic dimensions were chosen to approximate sheet flow for investigation of cell aggregation, but underlying computational methods also work for iso-symmetric flow⁹.”

c. Panels C & D in Figure 2 show that the velocity profile for the septic blood is both blunter and at a higher velocity than the control blood. As shown the septic blood appears to have a substantially higher flowrate. Please explain how this is possible if both types of blood samples are being delivered by syringe pump at the same flow rate. Was the video captured far enough downstream of the entrance of the channel to ensure a fully developed flow profile for both samples? Was the focal plane for the video capture set to the same depth into the channel, e.g 5 μm from the wall or to the depth with maximum velocity? Or is there a systematic error in the optical flow calculation related to the blood sample type (individual cells vs cell aggregates)?

- Experimental conditions were identical for both sepsis and control patient samples (e.g., the same amount of time elapsed from the time perfusion was started to the time the videos were collected and the very end of the channel region of interest was used for video capture). As discussed above, **we suspect aggregation does play a role in the observed changes in rheological properties, which would contribute to a substantially higher flow rate in sepsis.**
- The primary purpose of this manuscript is to introduce a new tool to a series of users by introducing each application and providing an experimental case study that demonstrates inputs, parameters, outputs, and potential use cases. **As such, our case studies in this manuscript are designed to give readers a sense of how the software can enable their experiments but are not meant to be the scientifically conclusive.** The full realization of this conclusion is outside of the intended scope of the iCLOTS project and will be explored in a future manuscript.

d. What is the spatial resolution of the optical flow method across the channel for the window size used? Supplement Fig 4 Panel B shows velocity profiles for different window sizes. Regardless of the window size in the dimension across the channel (h), there is a velocity value at +35 and -35 μm (supposedly at the walls). Explain how the optical flow algorithm can calculate velocity at the wall regardless of the window size. Give the window size in μm as well as pixels.

- We thank the Reviewer for their concern, which we feel is related to the Reviewer’s previous concerns with general usability, and as such our users would certainly benefit from clarification. As with all applications, **users select parameters based on their unique dataset.** The interactive analysis window of the velocity application (with a sample video selected for analysis) shows users a visual representation of each parameter value:

Colors for each parameter (left) match with the colors of the boxes and lines overlaid on the original imaging data (center) such that users have a visual sense of how the parameters they choose relate to their unique dataset. Each parameter is explained in detail in the in-software and online help guide/documentation. As users adjust parameters, the overlaid boxes and lines representing parameters also change in real time.

- **Spatial resolution for all iCLOTS applications is single pixel** (e.g., features to track can be found in any region of the image, including the wall). The iCLOTS velocity application exports the final frame, velocity value, and final pixel location of every feature tracked as a .csv file. Final profile values are given for a bin (in Figure 4B, a 2.5 µm-wide bin automatically calculated as the width of a linearly-spaced bin from the *n bins* parameter).
- To account for the wide variability of datasets iCLOTS might encounter, we ask the user to give a µm-to-pixel ratio specific to their microscopy setup. Final reported dimensions and velocities are reported with both pixels and µm as units. The formula used to convert from pixels to µm is:

$$\mu m = pix * \frac{1 \mu m}{n pix}$$

e. Finally, there is usually a relatively cell free plasma layer at the wall of blood vessels of this size due to migration of RBCs from higher shear rate at wall to lower shear near centreline and due to RBC collision with the wall. In Fig 4 Panel B there is no evidence to suggest a pseudo cell free layer at the edges of the channel and some RBCs appear to be cut-off suggesting the channel is wider.

- **We have found that in microfluidics, particularly in sheet-like devices such as the devices used here (please see point .b, above), the cell free layer is not always apparent, so as such, we consider this a limitation of these experiments.** However, as shared in the updated Figure 4B (please also see above), improved images showing the channel walls increase manuscript clarity.

“Blood was perfused using syringe pumps into devices for 28 minutes.” Why was a syringe pump used for the occlusion and accumulation assay rather than constant pressure? As the channels become occluded the resistance across the microfluidic device increases. Hence, the driving pressure across the channels increases as the syringe pump forces a constant flow through the device (same argument as above). This dynamically

changes the conditions for adhesion and occlusion over the 28 minute sample. With multiple channels the flow distribution among channels will also change due to the heterogeneous distribution of accumulation (as shown in Fig 6). Provide an argument for why a constant flowrate system was used and include the limitations.

- As shared above, syringe pump driven flow was primarily used for the occlusion and deformability experiments for simplicity and ease of use. Small sample sizes also limited the use of larger pressure control systems. **However, the effect of heterogeneous distribution of accumulation is a potential limitation, so at the reviewer's suggestion we have indicated this in the Supplementary methods for the occlusion/accumulation assays:**

“Blood was perfused using constant flow via syringe pumps (PhD Ultra, Harvard Apparatus) into devices for 28 minutes. In branching microfluidic devices such as the device presented here, flow distribution may change over time due to the heterogeneous distribution of accumulation.”

- Select experiments (e.g. quantification of blood velocity in deoxygenated sickle cell disease whole blood samples, Figure 5A) were performed using pressure-driven flow, indicating iCLOTS analysis is suitable for either syringe pump or constant pressure perfusion.
- **We thank Reviewer 4 for their concerns about the practical use of iCLOTS. With their guidance, we have updated the manuscript, supplement, and online and in-software user guide/documentation to provide better guidance for iCLOTS users, including selecting experimental methods, choosing appropriate parameters, and interpreting results.**

Concerns from Reviewer 5

This paper presents iCLOTS, a toolkit available for free download. iCLOTS includes four categories of applications: cell adhesion, single-cell tracking, velocity profile, and multiscale microfluidic-centric applications. iCLOTS is open source, easy to use and requires no coding expertise to implement. This allows iCLOTS to be used in different experiments. New observations are also reported in this paper benefiting from the efficiency of the iCLOTS analysis. The iCLOTS is novel and interesting, and their experimental evaluation is in general convincing.

- We thank the Reviewer for their comments on the novelty and potential for wide interest in iCLOTS, as well as stating that our evaluation is convincing.

However, there are some minor revision for the authors:

1. In Case study 3, the authors found that differences in adhesion to collagen-coated surfaces, platelet morphology, and phosphatidylserine (PS) exposure in platelet samples from patients with an FLI-1 mutation and HPS as compared to healthy controls. But there is no corresponding 'Figure 4F' in Figure 4 provided by the authors. There is also some ambiguity in the author's expressions. Does 'Supplementary Figure 6' correspond to the missing 'Figure 4F' or is it a comparison diagram for 'Figure 4F'? Supplementary Figure 6 provides several figures. Which one belongs to Case study 3?

- We thank the reviewer for noting that some figure references are improperly labeled. “Figure 4F” should refer to figure “5F”, and we have added the specific subfigure (6A) to our reference to Supplementary figure 6. Supplementary figure 6 is now Supplementary figure 7 after the addition of new figures.

2. In the second paragraph of the iCLOTS workflow section, the authors propose that iCLOTS can analyze data quickly, with processing times ranging from a few seconds to a few minutes, depending on the size and number

of files, but do not provide specific data proofs. In some of the images, the authors mark the time of analysis, but do not specify it. When working with very large datasets, can iCLOTS also limit the analysis time to a few minutes? Please provide the specific data quantities and the corresponding analysis time.

- In response to the Reviewer’s concerns about image processing times, we have added the requested data proofs using new supplementary text and table to a new **Software use** section to address these and other concerns:

Typically, after users adjust necessary parameters, analyses take on the order of seconds to minutes to run. To reduce processing time, users may resize or shorten files using the suite of video processing tools.

Application	Smaller dataset	Larger dataset
Fluorescence microscopy adhesion	One 60x 62 KB image	<1 second
Fluorescence microscopy filopodia count	One 60x 62 KB image	<1 second
Single cell tracking	5.1 MB video with ROI selected	13 seconds
Velocity profile	900 KB video	1 minute, 53 seconds
Microchannel accumulation	One tile scan 1.6 MB image	<1 second

“Supplementary table 1. iCLOTS analysis run times depending on application chosen and file size. After the user takes time to adjust parameters to their own specific dataset, analysis of the dataset takes on the order of seconds to minutes, with larger video files requiring the most time. Parameters and ROI chosen may affect analysis time. Exporting data may take additional time, particularly if a large number of labeled video frames are being exported. Tests were performed on a 2019 MacBook Pro with 16 GB RAM. Individual processing times may vary based on computer model and specifications.”

- These timing experiments for real-world data sets, as suggested by the Reviewer, will increase user understanding of practical use of the software.

3. In the second paragraph of the Discussion section, the authors suggest that blood samples are the most useful test benchmark, but do not give a detailed explanation and experimental proof. Please add an experiment to clarify it.

- In the manuscript introduction, we share our rationale for benchmarking the capabilities of our software using blood:

“Here, we present each application as applied to blood cells, the prototypical biofluid/biospecimen, which are subject to unique requirements and constraints including

heterogenous cell types, high cell densities, frequent integration of fluid flow, and increased viscosity.”

- We add in the manuscript discussion:

“As the prototypical biofluid biospecimen, we have found blood cells and related cell suspensions to be the most useful test case to benchmark iCLOTS for dense numbers of different cellular subpopulations, potentially under physiological flow conditions. As such, clearing a “high bar” with blood samples indicates that iCLOTS is also compatible with other biofluid/cell samples, which typically will be simpler than blood.”

iCLOTS algorithms are designed based on principles of feature and particle detection, which rely simply on patterns of pixel values, and are therefore not particular to blood cells.

- However, we wholeheartedly agree with the Reviewer that iCLOTS use with cell types other than blood would significantly increase the value of our software, and **to address the Reviewer’s concerns about iCLOTS use with other cell types, we have added two additional experiments analyzing data of other cell types and multi-cellular structures** within the new Supplementary figure 13, which recreates results of past *Nature Communications* articles from their Supplementary movie data.

- We share details of successful analysis of fibroblasts and coral polyps within the figure caption:

Supplementary figure 13. Data generated from iCLOTS recapitulates key findings from videomicroscopy of microfluidic experiments taken from previously published *Nature Communications* publications. (A) The iCLOTS single cell tracking application counts cells in motion as they move away from (control solution) or towards (chemokine solution) a chemokine. As published in *Boneschansker et al., 2014*, using Supplementary movie data downloaded freely from the *Nature Communications* website, we also find leukocytes in suspension migrate preferentially towards chemoattractant fMet-Leu-Phe (fMLP), that complement component 5a (C5a) induces both chemoattraction and repulsion in equal proportions in leukocytes, and that stromal cell-derived factor 1 (SDF-1) primarily repulses lymphocytes. (B) iCLOTS brightfield microscopy adhesion application counts coral polyps over time within Supplementary movie data monitoring a coral-on-a-chip microfluidic system, also indicating potential for use with other cells and/or multicellular structures. As published in *Shapiro et al., 2016*, we also find a gradual increase in salinity applied over time causes coral polyps to separate from each other. (C) iCLOTS accumulation and occlusion application quantifies fluorescence microscopy

signal from Supplementary movie videomicroscopy data of a microfluidic device designed to recreate the circadian system. As published in Gagliano et al., 2021, we also find that by controlling the timing, period, and frequency of metabolic perturbations, circadian gene expression as detected by luminescence oscillates over time. (D) iCLOTS accumulation and occlusion application quantifies fibrin deposition shown in Supplementary movie videomicroscopy of a specialized microfluidic device. As published in Jain et al., 2016, we also find that fibrin deposition from whole blood onto a microfluidic surface follows a sigmoidal trend.

Supplementary figure 13B shows iCLOTS analysis recapitulates results from Shapiro et al. (*Nature Communications*, 2016) showing that coral polyps separate under salinity stress. Supplementary figure 13C shows iCLOTS analysis recapitulates results from Gagliano et al. (*Nature Communications*, 2021) showing that fibroblasts with a fluorescent reported gene display circadian rhythms.

- We have included a reference to these experimental findings using other cell types in the main manuscript:

“iCLOTS was designed around large data sets from multiple research groups and has been used to recreate key findings from previously published studies⁷⁵⁻⁷⁸ (Supplementary figure 11) in order to verify that the algorithms applied work consistently for a variety of experimental set ups, imaging parameters, and additional cell or multicellular structure types.”

- While we found blood to be a useful benchmark for validating iCLOTS as it incorporates computationally challenging events such as diverse cell types and fluid flow, **in response to the reviewer’s concern that blood may not be the most useful test case we have performed additional experiments using iCLOTS to analyze fibroblast and coral polyp data, verifying that iCLOTS can be used for any cell type, or even multicellular structures.**

REVIEWERS' COMMENTS

Reviewer #1 (Remarks to the Author):

The authors have adequately addressed comments. I am pleased to see the team takes feedback on usability to heart and commits to continuous improvement of their software.

Reviewer #2 (Remarks to the Author):

The authors have done a reasonable job at addressing previous comments. There are a few remaining issues to address prior to publication:

1) Filopodia are typically defined as protrusions mediated by certain molecular factors, such as myosin X and fascin. Not all cell protrusions are mediated by these molecular factors, and the authors primarily count protrusions without validating molecular markers. The authors can either alter terminology to call their metric "protrusions" or "filopodia-like protrusions", or they can stain for filopodia-specific markers for multiple cells and determine how many of the identified protrusions (based on their metric) are actually filopodia (e.g. co-localized with myosin X or fascin).

2) In supplementary figure 11, what are the units of the x-axis for (a) and (b)?

Reviewer #3 (Remarks to the Author):

The authors have submitted a lengthy report in response to the five reviewers and have modified the paper to include additional results. My first review was generally positive and I stand by this review regarding the new improved paper.

I raised specific concerns about the potential uses of iCLOT in clinical laboratories. The new paragraph added in the resubmitted paper is satisfactory.

The authors have added a clarification on the concept of robustness in response to my concern. And they have added a new Supplementary Figure 14 to illustrate the relevance of image quality, which was not clear in the first paper version. This figure is relevant showing examples of the system with real images of poor quality.

In short, I have no further comments, being in favor of publishing this paper.

Reviewer #5 (Remarks to the Author):

Dear Authers,

I believe this manuscript has addressed my previous questions and I would like to express my appreciation for the efforts of your team in revising the manuscript.

RESPONSE TO REVIEWERS' COMMENTS

We thank you for suggesting that our manuscript “iCLOTS: open-source, artificial intelligence-enabled software for analyses of blood cells in microfluidic and microscopy-based assays” should be accepted for publication in *Nature Communications*. We would like to thank you for your continued constructive review, time, and thoughtful consideration of our manuscript and we were very encouraged by your interest in our open-source software and associated experimental findings. **In response to the Reviewers' valuable feedback, we have addressed each remaining concern**, which has continued to improve the quality of our manuscript and has made our software a more accessible and useful product for our target audience, the larger community of biomedical researchers and clinicians.

Below we address all remaining concerns from each individual reviewer, indicated with blue bolded text. All original reviewer comments are underlined. Responses are given in bulleted lists below the comment they directly address. “*Relevant text from the manuscript or supplement is in italics with revised changes in yellow highlight.*” **Key components of our response are bolded.**

Concerns from Reviewer 1

The authors have adequately addressed comments. I am pleased to see the team takes feedback on usability to heart and commits to continuous improvement of their software.

- We thank the reviewer for their comments about our commitment to usability and continuous improvement of our software. As we share in the manuscript:

“iCLOTS is shared with the greater research community with dedicated channels for feedback such that over time, as researchers contribute their own needs and workflows, the software will continue to develop and improve.”

We welcome all user feedback through our dedicated website contact page, <https://www.iCLOTS.org/contact> and/or via our github account, <https://www.github.com/iCLOTS>.

Concerns from Reviewer 2

The authors have done a reasonable job at addressing previous comments.

- We thank the reviewer for both the previous comments, which undoubtedly improved the manuscript, and for stating that we have adequately addressed them.

There are a few remaining issues to address prior to publication:

1) Filopodia are typically defined as protrusions mediated by certain molecular factors, such as myosin X and fascin. Not all cell protrusions are mediated by these molecular factors, and the authors primarily count protrusions without validating molecular markers. The authors can either alter terminology to call their metric “protrusions” or “filopodia-like protrusions”, or they can stain for filopodia-specific markers for multiple cells and determine how many of the identified protrusions (based on their metric) are actually filopodia (e.g. co-localized with myosin X or fascin).

- We agree with the reviewer that this is an important distinction. In the main manuscript, supplement, and all main manuscript and supplementary figures, we have altered our terminology to call our metric “protrusions” or “filopodia-like protrusions,” as suggested. For example, from the main manuscript:

“iCLOTS includes a single cell-resolution protrusion-counting tool (Figure 5D) based upon Harris corner detection, demonstrated here to count filopodia-like protrusions within platelets from healthy and clinical samples. Using this application, researchers can objectively apply criteria for protrusion detection (Supplementary figure 7).”

2) In supplementary figure 11, what are the units of the x-axis for (a) and (b)?

- Supplementary figures 11 (a) and (b) show the ratio of platelet (CD41+) or white blood cell (CD45+) signal from similar experiments using a clinical sample with and without crizanlizumab treatment. The figure has been corrected to indicate that the units are arbitrary.

Concerns from Reviewer 3

The authors have submitted a lengthy report in response to the five reviewers and have modified the paper to include additional results. My first review was generally positive and I stand by this review regarding the new improved paper.

- The authors thank the reviewer for their continued support.

I raised specific concerns about the potential uses of iCLOT in clinical laboratories. The new paragraph added in the resubmitted paper is satisfactory.

- As the authors are interested in eventually expanding versions of iCLOTS into clinical use, we appreciated this feedback and were happy to address the issue further.

The authors have added a clarification on the concept of robustness in response to my concern. And they have added a new Supplementary Figure 14 to illustrate the relevance of image quality, which was not clear in the first paper version. This figure is relevant showing examples of the system with real images of poor quality.

- As with any new tool, users may be hesitant to trust results or may struggle to adapt to its use. Both explaining the process by which we've developed robust methods and results and including easily interpretable examples of image quality were excellent suggestions by the reviewer that have undoubtedly made our product more usable by a wide audience.

In short, I have no further comments, being in favor of publishing this paper.

- The authors thank the reviewer for their support.